# Dynamic relocalization of cytosolic type III secretion system components prevents premature protein secretion at low external pH

Stephan Wimmi [1], Alexander Balinovic[2,6], Hannah Jeckel [2,3], Lisa Selinger[1], Dimitrios Lampaki[1,7], Emma Eisemann[1,8], Ina Meuskens[4], Dirk Linke [4], Knut Drescher [2,3], Ulrike Endesfelder [2,6] & Andreas Diepold [1,5✉]

Many bacterial pathogens use a type III secretion system (T3SS) to manipulate host cells. Protein secretion by the T3SS injectisome is activated upon contact to any host cell, and it has been unclear how premature secretion is prevented during infection. Here we report that in the gastrointestinal pathogens *Yersinia enterocolitica* and *Shigella flexneri*, cytosolic injectisome components are temporarily released from the proximal interface of the injectisome at low external pH, preventing protein secretion in acidic environments, such as the stomach. We show that in *Yersinia enterocolitica*, low external pH is detected in the periplasm and leads to a partial dissociation of the inner membrane injectisome component SctD, which in turn causes the dissociation of the cytosolic T3SS components. This effect is reversed upon restoration of neutral pH, allowing a fast activation of the T3SS at the native target regions within the host. These findings indicate that the cytosolic components form an adaptive regulatory interface, which regulates T3SS activity in response to environmental conditions.

[1] Department of Ecophysiology, Max Planck Institute for Terrestrial Microbiology, Marburg, Germany. [2] Max Planck Institute for Terrestrial Microbiology, Marburg, Germany. [3] Department of Physics, Philipps-Universität Marburg, Marburg, Germany. [4] Department of Biosciences, University of Oslo, Oslo, Norway. [5] SYNMIKRO, LOEWE Center for Synthetic Microbiology, Marburg, Germany. [6] Present address: Department of Physics, Mellon College of Science, Carnegie Mellon University, Pittsburgh, PA, USA. [7] Present address: Max-Planck-Institut für Immunbiologie und Epigenetik, Freiburg, Germany. [8] Present address: James Madison University, Harrisonburg, VA, USA. ✉email: andreas.diepold@mpi-marburg.mpg.de

In order to increase survival chances in contact to eukaryotic host cells, both symbiotic and pathogenic bacteria have developed methods to influence host cell behavior. The type III secretion system (T3SS) injectisome is a molecular machinery used by various pathogenic bacterial genera, including *Salmonella*, *Shigella*, pathogenic *Escherichia*, *Pseudomonas,* and *Yersinia* to deliver molecular toxins—effector proteins—directly into the eukaryotic host cells. In this manuscript, T3SS refers to the virulence-associated T3SS. The common "Sct" nomenclature[1,2] is used for T3SS components; when specifically referring to components of a particular species, the species initials are used as a prefix, e.g., *Ye*SctD for *Y. enterocolitica* SctD. While the T3SS effectors differ among the different bacterial species[3,4], and can have different functions in modulating the cytoskeleton, invading and escaping host cells or endosomes, or inducing host cell death (see reviews for animal pathogenic bacteria[1,5–10] and T3SS-plant interactions[11], the structural proteins of the injectisome are highly conserved[12,13]. An extracellular needle is formed by helical polymerization of a T3SS-exported protein, and ends in a pentameric tip structure. At the proximal end, the needle is anchored by two multimeric membrane rings that span the outer and inner membrane. Additionally, the inner membrane (IM) ring encloses

the export apparatus. At the cytosolic interface of the injectisomes, four soluble T3SS components (SctK/Q/L/N) interact to form six pod structures[14,15] (Fig. 1a).

The soluble T3SS components SctK, SctQ, SctL, and SctN interact in a linear fashion[16], and the presence of all four proteins is needed for their assembly at the injectisome, and subsequently for effector secretion[17,18]. SctK/Q/L form a high molecular weight complex that has been shown to bind chaperones and effectors[19]. Since the cytosolic components do not co-purify with the rest of the needle, their structural arrangement has only been revealed by recent in situ cryo-electron tomography[14,15,20,21], which showed the formation of six pod structures at the cytosolic interface of the injectisome. The connection between these pod structures and the membrane rings is established by the cytosolic component SctK, which binds to SctD. In the presence of SctK, the cytosolic domains of SctD (SctD$_C$) rearrange and in the case of the *Salmonella* SPI-1 T3SS form six discrete patches of four SctD$_C$ each that interact with one SctK protein, which in turn connects to SctQ, SctL, and SctN[15,22,23].

In addition to forming the injectisome-bound pod structures, the cytosolic components exist in a freely diffusing cytosolic state, with proteins exchanging between the two states[24]. In the cytosol,

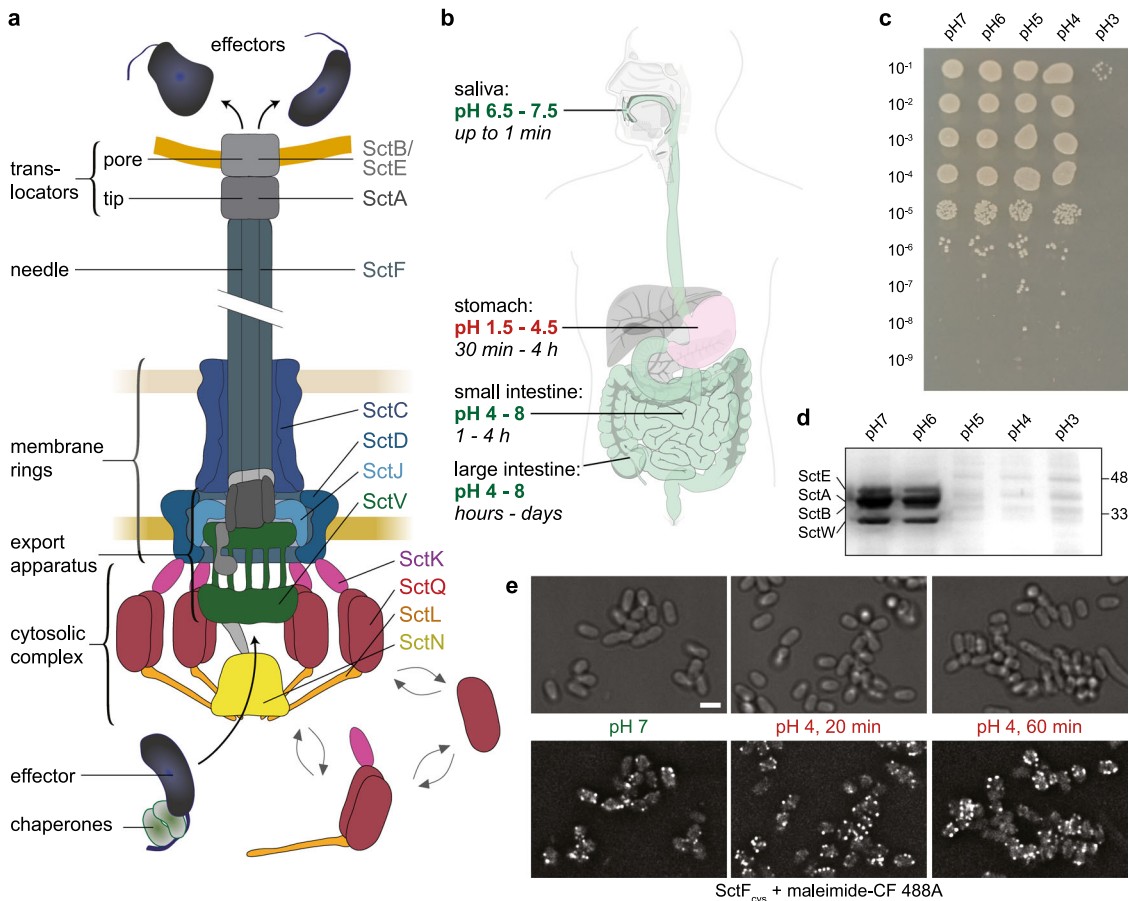

**Fig. 1 The *Yersinia enterocolitica* T3SS and its function and stability at low pH. a** Schematic representation of the active T3SS injectisome (modified from ref. [68]). Left side, description of the main substructures; right side, names of T3SS components studied in this manuscript. Double arrows indicate the exchange of the cytosolic subunits or subcomplexes between the cytosolic and the injectisome-bound state. **b** pH ranges and typical retention times at different parts of the gastrointestinal system[43, 44]. Digestive tract image based on the public domain template https://commons.wikimedia.org/wiki/File: Digestive_system_diagram_de.svg. **c** Dilution drop test of *Y. enterocolitica* cultures incubated at the indicated pH at 37 °C for 30 min. **d** In vitro secretion assay showing the export of native T3SS substrates (indicated on left side) in strain IML421asd at the indicated external pH values. Black and white scan of a coomassie-stained SDS-PAGE gel; supernatant of $3 \times 10^8$ bacteria per lane. Right side, molecular weight marker (kDa). Secretion assay of strain containing effectors, see Supplementary Fig. 3a. **e** Staining of T3SS needles at the different indicated external pH values; time indicates the duration of pH 4 treatment. A strain expressing the mutated needle subunit SctF$_{S5c}$ was covalently labeled with maleimide-CF 488A. Experiments for **c**, **d**, and **e** were performed at least 3 times with comparable results. Scale bar, 2 μm.

SctK/Q/L/N form a dynamic adaptive network, with a variety of complexes of different stoichiometries that change their composition in response to different external conditions[18,25]. Strikingly, protein interactions and exchange rates amongst the cytosolic components correlate with the activity of the injectisome[18,24,26]. However, the precise role of the cytosolic components in the secretion process remains unclear to this date.

*Yersinia enterocolitica* is an extracellular gastrointestinal pathogen that employs its T3SS to downregulate immune responses and prevent inflammation after the penetration of the intestinal epithelium. For initial attachment to the epithelium, the bacteria employ a number of adhesins[27,28]. The two major adhesion factors at the early stage of infection are the autotransporters invasin (inv), which is already expressed at ambient temperatures[29,30] and *Yersinia* adhesin A (YadA)[31]; *inv yadA* double mutants are completely avirulent[28,32]. YadA is a key factor for establishing an infection[33], which interacts with a variety of extracellular matrix molecules, including collagen and fibronectin[34]. YadA establishes close contact to the host cells enabling injection; its length is tightly linked to the length of the injection needle of the T3SS[35]. *Y. enterocolitica* is usually taken up with contaminated food or water. The shift from environmental to host temperature (37 °C) induces expression and assembly of the injectisomes as well as YadA[36,37]. The bacteria then have to pass the acidic environment of the stomach. During that time, the injectisome can already be present and ready for secretion of translocators (which form a pore in the host membrane and its connection to the needle, Fig. 1a) and effectors. Since the T3SS readily translocates cargo into any host cell type it adheres to, including immune cells, epithelial cells, and even red blood cells[38–42], a distinct mechanism is needed to prevent premature activation of the T3SS during the passage of low pH parts of the gastrointestinal system, such as the stomach (external pH of 1.5–4.5[43,44], Fig. 1b), which would result in a loss of valuable resources, or even elicit immune responses.

We hypothesized that acidic environment may be detected by bacteria to inhibit effector secretion under these conditions. Here, we investigated this hypothesis with a combination of fluorescence microscopy, single-particle tracking, and functional assays. Our findings show that although parts of the T3SS, including the extracellular needle, are stable at low pH, a set of cytosolic T3SS components temporarily unbind from the injectisome at low external pH. The reversible dissociation corresponds to a temporary suppression of effector secretion by the T3SS. This mechanism prevents premature activation of the T3SS, while ensuring the reactivation of the T3SS, once a pH-neutral environment is reached. Our data provide a striking example for how bacteria apply protein dynamics to adapt the function of a large macromolecular machine essential for virulence to external conditions.

## Results

**Resistance of *Y. enterocolitica* and its T3SS needles to low external pH.** To investigate to which degree *Y. enterocolitica* can withstand a drop in external pH, *Y. enterocolitica* cultures in the exponential growth phase were exposed to different pH for 30 min at 37 °C. A subsequent dilution series on neutral pH agar showed high survival down to pH 4 (Fig. 1c). Similarly, while the optical density of bacterial cultures slightly decreased over 90 min at pH 4 at 37 °C, growth recovered at a reduced rate upon restoration of neutral pH (Supplementary Fig. 1). Together, these results show that *Y. enterocolitica* tolerates temporary incubation down to pH 3, with limited fitness decrease at pH 4 and above. *Y. enterocolitica* adheres to host cells, other bacteria and abiotic surfaces by adhesins, most importantly the trimeric adhesin YadA

and invasin[31,34,45]. We thus tested whether low pH prevents the binding of YadA to collagen and more generally, of *Y. enterocolitica* cells to surfaces. Binding could be established at low pH in both cases, albeit at a significantly reduced level (Supplementary Fig. 2, Supplementary Movies 1–2). These results suggest that to avoid protein translocation into non-host cells, secretion itself might be prevented at low pH.

To test under which conditions *Y. enterocolitica* secretes T3SS substrates, we performed an in vitro secretion assay where *Y. enterocolitica* cells primed for secretion were subjected to secreting media in the range from pH 8 to pH 4. Indeed, we observed that secretion did not occur at low pH (Fig. 1d, Supplementary Fig. 3a). This lack of secretion is not due to lower protein synthesis at pH 4 (Supplementary Fig. 3b), suggesting a specific mechanism to suppress secretion at low external pH.

But how does *Y. enterocolitica* prevent secretion at low pH? To determine if the absence of secretion is due to a complete disassembly of injectisomes at that pH, we visualized the needles at different pH values and over time by labeling an introduced cysteine residue with a maleimide-linked dye[46]. The needles were stable at pH 4 over continued time periods (Fig. 1e, Supplementary Fig. 4).

Taken together, *Y. enterocolitica* as well as its injectisome needles can withstand low external pH conditions, but still translocators and effectors are not secreted.

**Association of the dynamic cytosolic T3SS components to the injectisome is temporarily suppressed at low external pH.** We have recently found that the cytosolic T3SS components (SctK/Q/L/N) form a dynamic network, where protein exchange is connected to the function of the injectisome (Fig. 1a)[18,24]. Hence, we wondered whether these dynamic components could be involved in the inhibition of secretion at low pH. To investigate this question, we performed flow-cell-based total internal reflection fluorescence (TIRF) microscopy with functional N-terminal fluorescent protein fusions of the cytosolic components, expressed at native levels: EGFP-SctK, EGFP-SctQ, EGFP-SctL, and EGFP-SctN (Fig. 1a)[17,18]. At neutral or near-neutral pH, the cytosolic components localized in foci at the bacterial membrane, which represent their injectisome-bound state[17]. However, at an external pH of 4, all cytosolic components lost this punctuate localization, and the proteins relocated to the cytosol (Fig. 2a–c). The relocation remained stable over time at low external pH (Supplementary Fig. 5). Strikingly, this phenomenon was reversible: upon exposure to neutral external pH, the foci recovered within a few minutes (Fig. 2a–c). This effect was observed both under secreting and non-secreting conditions (Supplementary Fig. 6a), and was independent of the fluorophore or visualization tag that was used (Supplementary Fig. 6b). Notably, while the fluorescence of genetically encoded fluorophores such as EGFP is pH-dependent[47,48], the observed effect is not caused by the reduction of overall fluorescence in the cytosol (Fig. 2c, Supplementary Fig. 7).

Dissociation and re-association of the cytosolic T3SS components in response to the external pH was reversible for several cycles (Supplementary Movie 3), and dissociation kinetics were similar under non-secreting and secreting conditions (presence of 5 mM $CaCl_2$ and EGTA, respectively) (Supplementary Fig. 8). To quantify the kinetics of association and dissociation of SctQ, we monitored EGFP-SctQ foci in a microscopy flow cell after a pH change from 7 to 4 and vice versa. Foci gradually disappeared within 2 min under secreting conditions (Fig. 2d, Supplementary Fig. 9, Supplementary data 1, Supplementary Movie 4). Upon restoration of neutral external pH, this phenotype was reversed

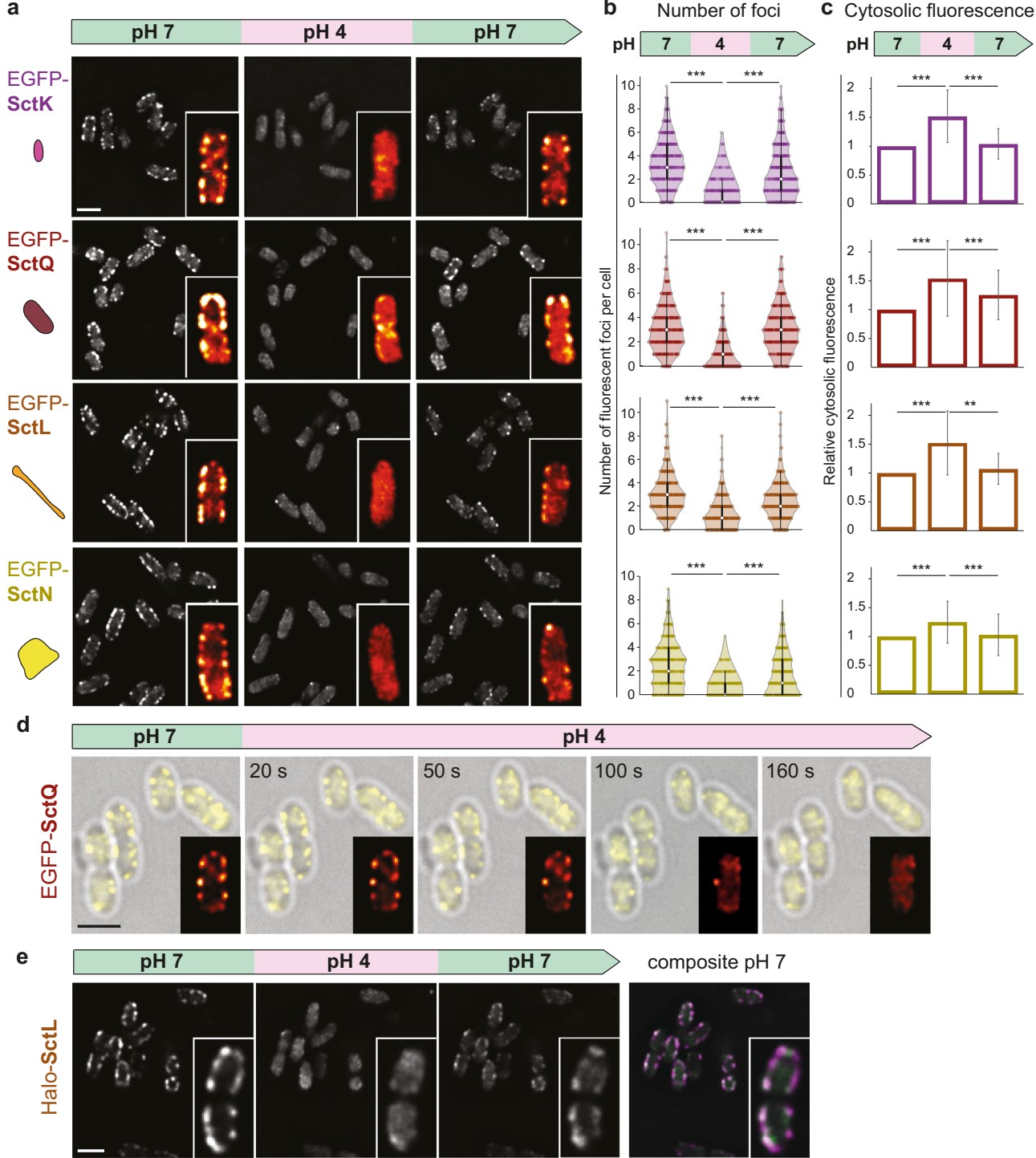

**Fig. 2 The cytosolic T3SS components temporarily dissociate from the injectisome at low external pH. a** Fluorescent micrographs of the indicated proteins in live *Y. enterocolitica*, consecutively subjected to different external pH in a flow cell. Images were taken under secreting conditions, 10 min after bacteria were subjected to the indicated pH. Insets, enlarged single bacteria, visualized with the ImageJ red-hot color scale. **b** Quantification of foci per bacterium for the strains and conditions shown in (**a**). $n = 453, 792, 466$, and 554 foci (from top to bottom) from three independent experiments. White circle denotes median, black bar denotes first and third quartile, black line denotes the lower/upper adjacent value. **c** Quantification of mean cytosolic fluorescence for the strains and conditions shown in (**a**). $n = 75$ cells from three independent experiments per strain. Bars denote mean values. For **b**, **c** error bars denote standard deviation; **/\*\*\***, $p < 0.01/0.001$ in homoscedastic two-tailed *t*-tests (**b**: $p \approx 2 \times 10^{-114}/2 \times 10^{-57}$, $4 \times 10^{-102}/1 \times 10^{-51}$, $5 \times 10^{-79}/1 \times 10^{-69}$, $8 \times 10^{-89}/2 \times 10^{-51}$; **c**: $p \approx 8 \times 10^{-18}/7 \times 10^{-13}$, $5 \times 10^{-11}/2 \times 10^{-3}$, $2 \times 10^{-13}/3 \times 10^{-9}$, $6 \times 10^{-8}/4 \times 10^{-4}$ for differences between pH for first/second pH 7 incubation, respectively, from top to bottom). **d** Kinetics of EGFP-SctQ dissociation after pH shift from 7 to 4. Overlay of phase contrast (gray) and fluorescence images (yellow); insets, enlarged single bacterium, visualized with the ImageJ red-hot color scale. **e** Fluorescence micrographs of Halo-SctL, labeled with JF 646 prior to the first image, during consecutive incubation at different external pH as in (**a**). Right, overlay of fluorescence distribution at pH 7 prior to and after pH 4 incubation (magenta and green, respectively). Experiments for **a**, **d**, and **e** were performed >5, >5, and 3 times, respectively, with comparable results. Scale bars, 2 μm.

and foci were restored within a similar time range (Supplementary Fig. 9, Supplementary Fig. 10, Supplementary Movie 5).

To test whether the recovery of foci was due to the synthesis of new proteins or if the previously bound proteins rebind to the injectisomes, we covalently labeled the pool of Halo-tagged SctL with the Janelia Fluor JF 646 Halo ligand dye[49] prior to the incubation at pH 4 to ensure that only the protein pool present at the time of labeling was fluorescent. Similar to the previous experiments, we observed a reversible loss of fluorescent foci at the membrane and an increase in cytosolic signal at pH 4 and rebinding of the fluorescent proteins upon changing pH to 7 (Fig. 2e). The cytosolic components colocalize with the needle at the initial pH 7 (Supplementary Fig. 11) and an overlay of the micrographs at pH 7 before and after the incubation at pH 4 indicated that most foci reform at the same position as they previously appeared (Fig. 2e, Supplementary Fig. 6b, Supplementary Fig. 11).

Taken together, our data indicate that the cytosolic components reversibly dissociate from the injectisome at low external pH. Upon exposure to neutral pH, proteins from the same pool rebind at the cytosolic interface of the injectisome, forming the potential basis for a regulatory mechanism for the prevention of secretion at low pH.

**Molecular mechanism of extracellular pH sensing.** What is the molecular basis for the dissociation of the cytosolic T3SS

components at low external pH? To find out whether the pH is sensed intracellularly, we first tested the impact of the changed external pH on the cytosolic pH, using a ratiometric pHluorin GFP variant (pHluorin$_{M153R}$[50,51]) as a pH sensor. Upon changing the external pH from 7 to 4, the cytosolic pH dropped to a mildly acidic value (pH 6.3–6.4). This cytosolic pH was retained for at least 30 min at external pH 4, but quickly recovered upon re-establishment of neutral external pH (Fig. 3a).

To test if this mild drop in cytosolic pH directly causes the dissociation of the cytosolic complex, we treated bacteria with the proton ionophore 2,4-dinitrophenol, which attunes the cytosolic pH to the external pH[52–54] (Supplementary Fig. 12), and visualized the localization of EGFP-SctQ at different pH values. EGFP-SctQ remained localized in foci representing assembled cytosolic complexes at pH 6.3 and 6.0 (Fig. 3b), indicating that the observed disassembly of the cytosolic complex is not caused by the mild acidification of the cytosol.

Based on the above results, we searched for a periplasmic pH sensor. To this aim, we tested the effect of overexpression of the four candidate proteins that have periplasmic domains and influence the localization of the cytoplasmic components—SctC, SctD, SctJ, and SctV (Fig. 1a). We reasoned that if pH-induced partial dissociation of one of these components induces the dissociation of the cytosolic complex, overexpression of this component would shift the equilibrium toward the associated form of the protein itself and, in consequence, decrease

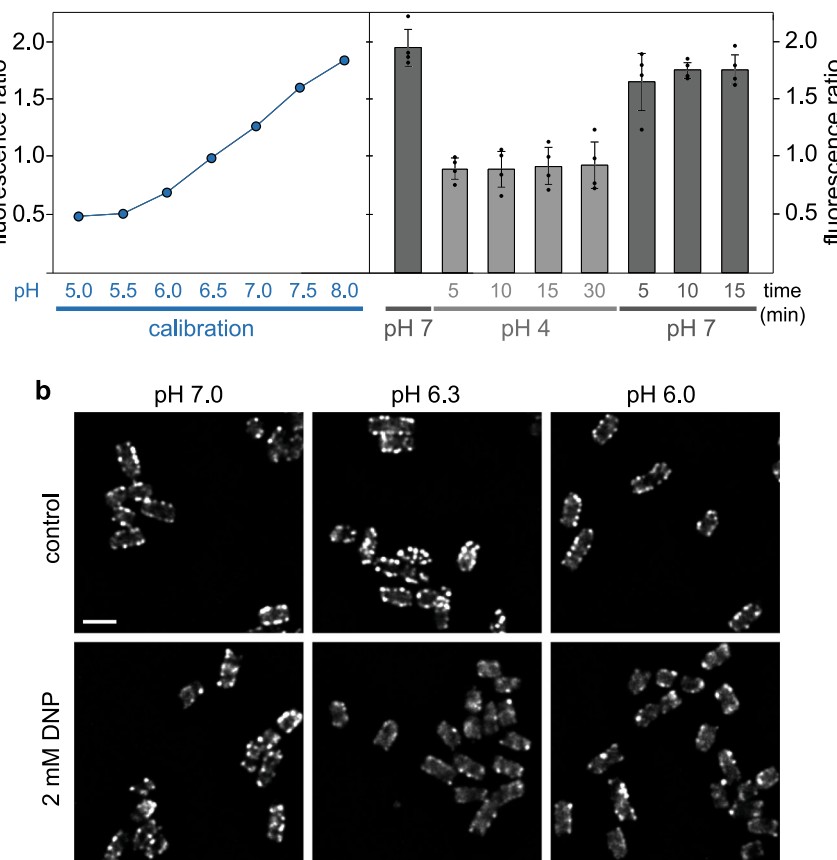

**Fig. 3 Low external pH leads to a small drop in cytosolic pH, which does not induce the dissociation of cytosolic T3SS components. a** Left, calibration of (Ex$_{390nm}$/Ex$_{475nm}$) fluorescence ratio of purified pHluorin$_{M153R}$ for the indicated pH values. Technical triplicate, error bars too small to display. Right, determination of cytosolic pH upon changing the external pH from 7 (first column) to 4 (columns 2–5) and back (columns 6–8). Fluorescence ratio (Ex$_{390nm}$/Ex$_{475nm}$) of bacteria expressing cytosolic pHluorin$_{M153R}$. $n = 4$ independent experiments, single data points indicated by small circles. Bars denote mean values, error bars denote standard deviation. **b** Fluorescence distribution of EGFP-SctQ in live *Y. enterocolitica* at indicated external pH in absence (top) or presence (bottom) of the ionophore 2,4-dinitrophenol (DNP). The experiment was performed 3 times with comparable results. At pH 4.0, no foci are present. Scale bar, 2 μm.

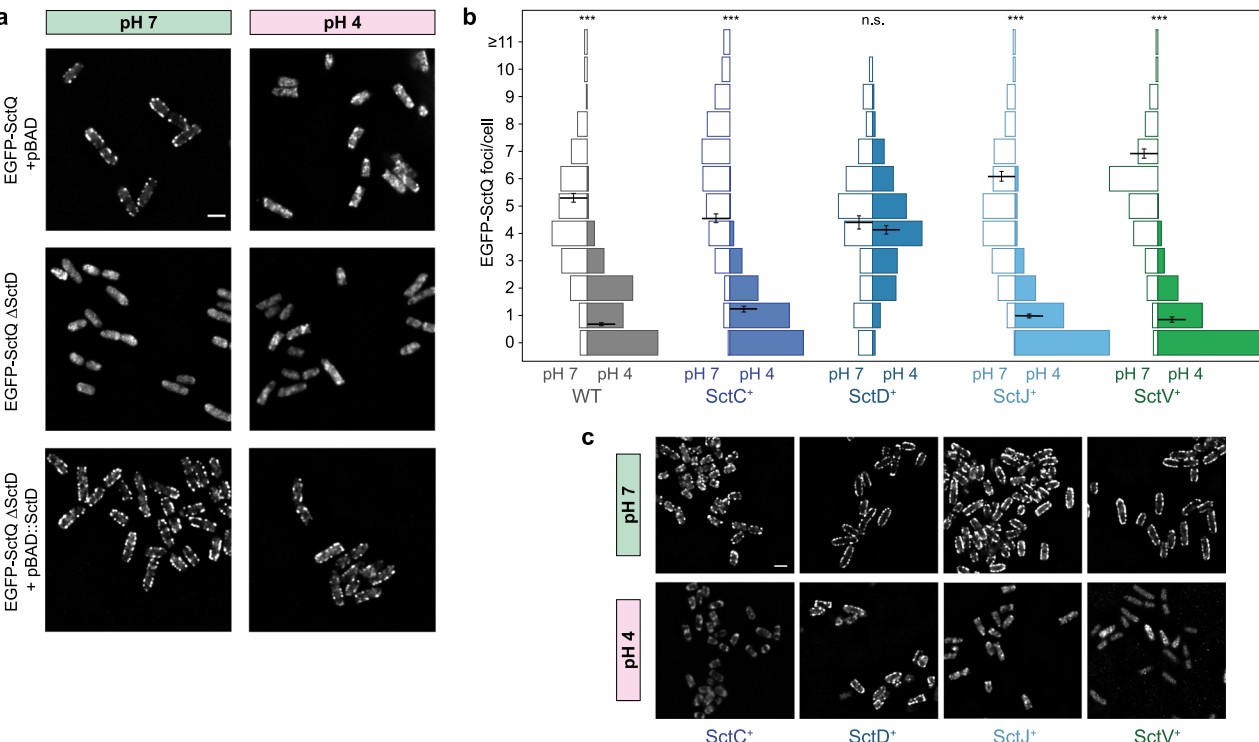

**Fig. 4 Overexpression of SctD, but not of SctC, SctJ or SctV suppresses the dissociation of the cytosolic components at low external pH. a** Cellular localization of EGFP-SctQ in live *Y. enterocolitica* at the indicated external pH. Top, wild-type; center, ΔSctD; bottom, additionally overexpressing SctD from plasmid. $n = 3$ independent experiments. **b** Quantification of EGFP-SctQ foci per bacterium in wild-type EGFP-SctQ strain and strains overexpressing the indicated proteins (induced by 0.2% arabinose), at external pH 7 (left, open) and pH 4 (right, filled). Black lines and error bars indicate mean number of foci and standard error of the mean. The number of foci at pH 7 and pH 4 does not differ significantly in strains overexpressing SctD (n.s., $p = 0.32$ in a homoscedastic two-tailed t-test), but in all other strains (***$p \approx 2 \times 10^{-126}$, $3 \times 10^{-43}$, $2 \times 10^{-112}$, $3 \times 10^{-106}$ for WT and overexpression of SctC, SctJ, and SctV, respectively). Bars display fraction of cells; $n = 199/310$, $203/149$, $115/137$, $205/279$, $178/184$ cells per strain and condition from 9/7, 8/10, 8/10, 9/9 8/9 fields of view (pH 7/pH 4 from left to right) from three independent experiments. **c** Cellular localization of EGFP-SctQ in the strains analyzed in (**b**). All tested strains show normal secretion, see Supplementary Fig. 14. Scale bars, 2 μm.

dissociation of the cytosolic components at low pH. Indeed, overexpression of SctD strongly reduced the dissociation of the cytosolic component EGFP-SctQ at low external pH in an induction-dependent manner—while at wild-type SctD levels the number of EGFP-SctQ foci was strongly reduced at pH 4, bacteria with increased levels of SctD retained a higher number of EGFP-SctQ foci at pH 4 (Fig. 4, Supplementary Fig. 13a–c). In contrast, overexpression of SctC, SctJ or SctV did not interfere with pH-induced dissociation of the cytosolic components (Fig. 4b, c, Supplementary Fig. 14). Taken together, these results strongly support a central role for SctD in the reaction to low external pH.

SctD is a bitopic IM protein connecting the outer membrane (OM) ring to the cytosolic components (Fig. 1a)[15,55], that is an obvious candidate to act as a sensor for the external pH. Earlier studies showed that lack of SctD, or its inability to bind to SctC, leads to a similar cytosolic location of SctK/L/N/Q as observed at an external pH of 4 (Fig. 2a)[17,18], and that the cytosolic domains of SctD connect to SctK via specific interactions (with four SctD binding to one SctK in *Salmonella* SPI-1)[15,22,23]. This suggests that structural rearrangements of SctD could lead to the dissociation of SctK and, subsequently, all other cytosolic components.

To investigate this hypothesis, we tested the behavior of SctD at low external pH by analyzing the localization of SctD. An EGFP-SctD fusion is stable and shows a slightly reduced effector secretion (ref. [24], Supplementary Fig. 15a, b). At pH 4, EGFP-SctD foci in the membrane became less clearly defined than at pH 7, with a concomitant increase in fluorescence throughout the

membrane (Fig. 5a). In contrast to the cytosolic components, however, the SctD foci did not completely disappear, suggesting a reduced affinity at low external pH. Importantly, like the cytosolic components, SctD recovered its localization in foci at neutral external pH within minutes (Fig. 5a). To study this unique phenotype in more detail, we performed single-molecule tracking of PAmCherry-SctD proteins in photoactivated localization microscopy (PALM). These experiments revealed that at an external pH of 7, more than 90% of the SctD molecules in the IM were static (medium jump distance (mjd) of 0–60 nm, corresponding to the measurement uncertainty[56,57]); by contrast, at an external pH of 4, more than 40% of the SctD molecules became mobile within the membrane (Fig. 5b, Supplementary Fig. 16), indicating a dissociation from the core injectisome structure at low external pH. In line with these findings, the large T3SS export apparatus component SctV-EGFP retained its localization within foci at pH 4; however, these foci (likely representing the SctV nonamers) became mobile within the membrane (Fig. 5c, Supplementary Fig. 17). The same behavior has been detected for SctV-EGFP in the absence of SctD[58], supporting the notion that at an external pH of 4, SctV is released from the SctD structure. To test whether the N-terminal fluorophore tag influences the stability of the IM ring, we localized EGFP-SctQ, which depends on the presence of the IM ring for its localization[17], at different external pH. The fusion of mCherry to SctD did not influence the formation of EGFP-SctQ foci in any tested condition (Supplementary Fig. 15c), suggesting a labeling-independent effect of external pH on SctD assembly.

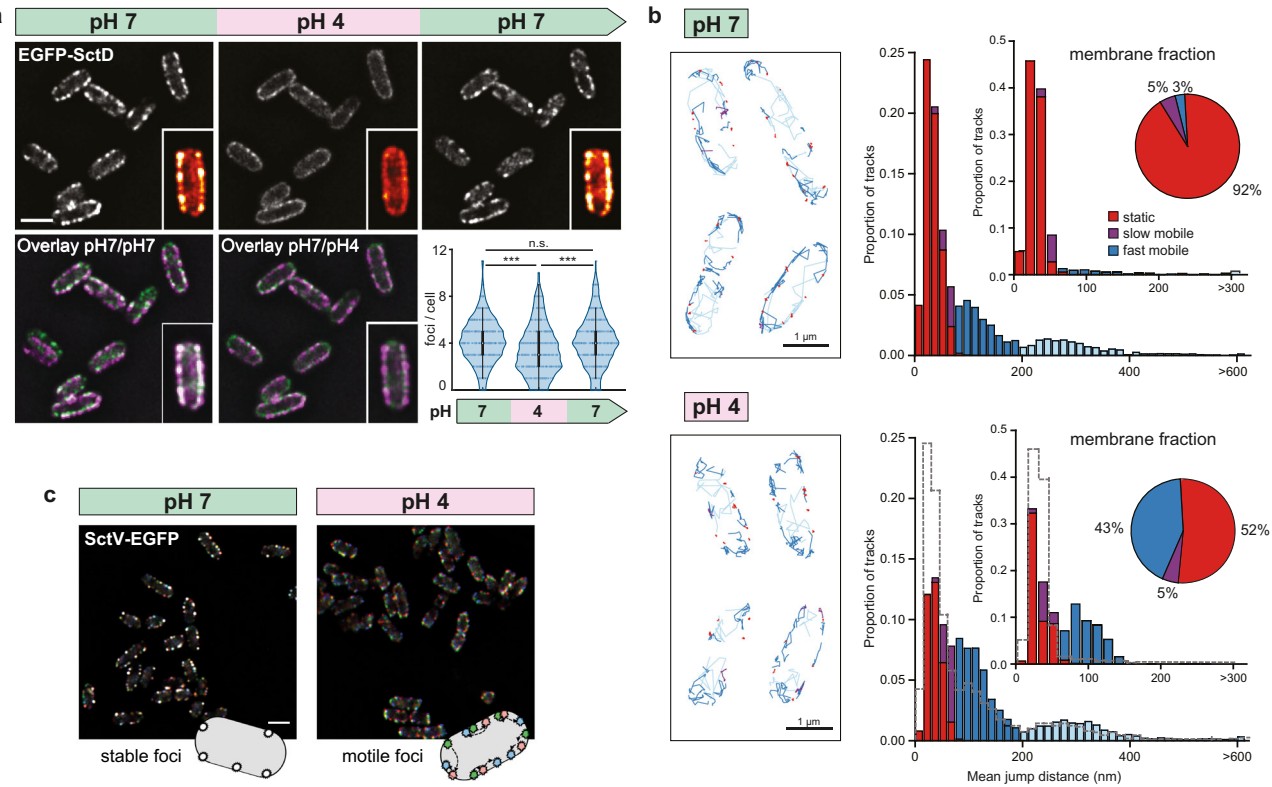

**Fig. 5 The bitopic IM protein SctD reacts to low external pH. a** Fluorescent micrographs of EGFP-SctD in live *Y. enterocolitica*, consecutively subjected to different external pH in a flow cell. Images were taken under secreting conditions, 10 min after bacteria were subjected to the indicated pH. Insets, enlarged single bacteria, visualized with the ImageJ red-hot color scale. Bottom, overlays of fluorescence at pH 7 before pH change (magenta) and pH 7 after pH change or pH 4 (green). Bottom right, quantification of foci per bacterium. $n = 468$ cells from three independent biological replicates. White circle denotes median, black bar denotes first and third quartile, black line denotes the lower/upper adjacent value. ***$p < 0.001$ in homoscedastic two-tailed *t*-tests; n.s., difference not statistically significant ($p \approx 2 \times 10^{-11}$, $2 \times 10^{-8}$, 0.42 for pH 7/4, pH 4/7 and pH 7/7, respectively). **b** PAmCherry-SctD dynamics in exemplary living *Y. enterocolitica* (left) and histograms (right) of the mean jump distances (MJD) of PAmCherry-SctD trajectories weighted by the number of jump distances used for calculating each MJD. Only trajectories with more than 6 one-frame jumps are shown and included into the analysis. Upper panel was measured at pH 7, lower panel at pH 4. Trajectories are assigned into two diffusive states: static (red) and mobile (violet and blue fractions) based on the experimental localization precision. Mobile trajectories are sorted into three MJD categories: lower than 60 nm (violet), lower than 195 nm (dark blue) and higher than 195 nm (light blue). Counts were normalized to the total number of trajectories. At pH 7 we acquired 29,859 trajectories (60% static, 2% mobile <60 nm MJD, 25% mobile from 60 to 195 nm MJD, 13% mobile >195 nm MJD; 19,473 membrane-bound trajectories). At pH 4 we acquired 33,036 trajectories (34% static, 4% mobile <60 nm MJD, 44% mobile from 60 to 195 nm MJD, 18% mobile >195 nm MJD; 21,600 membrane-bound trajectories). The bin size of 15 nm was calculated using the Freedman-Diaconis rule. Inset histograms and pie plot display statistics of only membrane-bound trajectories. The number of trajectories of membrane-bound static PAmCherry-SctD molecules decreases in pH 4 (92% to 52%), while the number of diffusing PAmCherry-SctD with MJDs <195 nm (dark blue bars) increases (3% to 43%). The fast fractions of PAmCherry-SctD molecules of MJDs >195 nm (light blue), only visible for the whole-cell analysis, remain constant. Gray dashed lines in the pH 4 panels represent the outlines of the pH 7 analysis. **c** Spatial stability of SctV-EGFP foci over time at the indicated external pH. Three images of the same focal plane were taken at 10 s intervals. Green/blue/red channel correspond to $t = 0/10/20$ s. See Supplementary Fig. 17 for single time points and larger fields of view. The experiment was performed three times with similar results. Scale bars, 2 µm.

## Physiological advantage of temporary suppression of type III secretion at low pH.

We reasoned that bacteria could benefit from the dissociation of the cytosolic T3SS components at low external pH to suppress secretion in the low pH parts of the gastrointestinal system, including the stomach, where pH values of 4 and below prevail, and where T3SS activation might lead to energy depletion or even elicit immune reactions. If this would be the case, bacteria with a similar infection route, but not bacteria that do not pass the gastrointestinal tract during normal infection, and therefore are not under evolutionary pressure to suppress T3SS activity at low pH, would be expected to display the same pH dependence for the localization of cytosolic T3SS components. We therefore tested the localization of cytosolic components at low external pH in *Shigella flexneri* and *Pseudomonas aeruginosa*. *S. flexneri* is a gastrointestinal pathogen that uses an evolutionarily distant T3SS, but like *Y. enterocolitica*,

needs to pass the stomach to invade the colonic and rectal epithelium. In contrast, the T3SS of *Y. enterocolitica* and *P. aeruginosa* are closely related[59], but the infection strategies of the two species differ. *P. aeruginosa* is not a gastrointestinal pathogen and mainly enters the host body through wounds. *S. flexneri* GFP-*Sf*SctN[60] formed fewer and less distinct foci than *Y. enterocolitica* EGFP-*Ye*SctQ and *P. aeruginosa* EGFP-*Pa*SctQ (Supplementary Fig. 18), possibly due to the lower stoichiometry of SctN compared to SctQ. Most importantly, in agreement with the hypothesis that the dissociation of cytosolic components at low external pH is a conserved adaptation to the passage of low pH parts of the gastrointestinal system during a normal infection, the fraction of *S. flexneri* GFP-*Sf*SctN with foci significantly decreased at pH 4, whereas *P. aeruginosa* EGFP-*Pa*SctQ cells showed a small, non-significant decrease (Fig. 6, Supplementary Fig. 18).

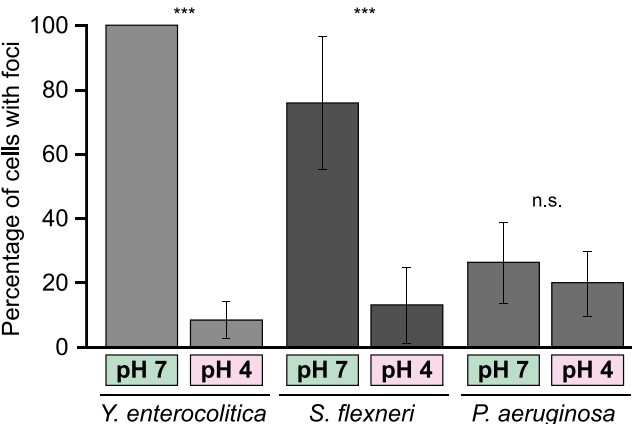

**Fig. 6 The effect of external pH on the assembly of cytosolic T3SS components is species-specific.** Percentage of bacteria with fluorescent foci at the indicated external pH for *Yersinia enterocolitica* EGFP-*Ye*SctQ, *Shigella flexneri* GFP-*Sf*SctN, and *Pseudomonas aeruginosa* EGFP-*Pa*SctQ, respectively. $n = 1134/1270$, 530/691, 1078/926 bacteria from 20, 22, 25 fields of view (pH 7/pH 4 from left to right) from three independent experiments per condition. Bars denote mean values, error bars denote standard deviation between fields of view. n.s., difference not statistically significant; ***$p < 0.001$ in a two-tailed homoscedastic $t$-test ($p \sim 2 \times 10^{-44}$, $6 \times 10^{-16}$, 0.053, respectively).

To determine a potential molecular mechanism for this effect, we looked for amino acids in the periplasmic portions of the T3SS that can be protonated at pH 4, but not at pH 7 (Asp, Glu, His) in *Y. enterocolitica* and *S. flexneri*, but not in *P. aeruginosa*. As no such amino acids could be identified (Supplementary Table 1), we focused on protonatable amino acids that differ between *Y. enterocolitica* and *P. aeruginosa* (Supplementary Fig. 19). Single amino acid substitutions, as well as the combination of all four substitutions still led to dissociation of the cytosolic component SctQ, as well as a loss of secretion at pH 4 (Supplementary Fig. 20). Notably, a *Y. enterocolitica* ΔSctD strain could be complemented for secretion by *in trans* expression of *Pa*SctD, which showed the same expression level-dependent suppression of the dissociation of the cytosolic components at low external pH as *Ye*SctD (Supplementary Fig. 13d–e). Taken together, these results and the clear effect of SctD overexpression (Fig. 4) suggest that the effect of the pH is conveyed by gradual rearrangement of intermolecular interactions rather than by discrete salt bridges.

In the more pH-neutral intestine, *Y. enterocolitica* need to pass the M cells across the intestinal epithelium. Afterward, the injectisome must be ready to manipulate immune cells. To test whether the reversible dissociation of the cytosolic T3SS components supports a fast activation of the T3SS once back at neutral pH, we monitored protein secretion over time after a temporary drop of external pH to 4. We found that secretion was suppressed at low pH, but restarted 20-40 min after reaching neutral pH (Fig. 7a). Notably, this recovery is much faster than the onset of effector secretion after de novo assembly of the T3SS by a temperature change to 37 °C (Fig. 7a), supporting the notion that bacteria benefit from the temporary dissociation of the cytosolic subunits at low pH in two ways: This mechanism suppresses protein secretion at low external pH, while ensuring a timely reactivation upon reaching a pH-neutral environment (Fig. 7b).

## Discussion
On their way through the gastrointestinal system, bacteria encounter a multitude of different pH environments. Importantly, the highly acidic stomach acts as natural barrier for food-borne infections. Gastrointestinal pathogens express factors that facilitate survival in these conditions, such as urease, of which high amounts are exported in *Y. enterocolitica*[61–65]. It was not known, however, if and how the activity of the T3SS, an essential virulence factor for many gastrointestinal pathogens, is regulated under these conditions. Although the T3SS target cells are downstream of the stomach for most gastrointestinal pathogens, cells in the low pH parts of the gastrointestinal system can be accessible and bacteria can attach to host cells at low pH (Supplementary Fig. 2, Supplementary Movies 1–2). Our data support the notion that bacteria prevent premature injection into host cells at this stage by directly using the external pH as a cue for the temporary suppression of the T3SS. While parts of the injectisome, including the needle, remain stable, the cytosolic components dissociate in an acidic environment (pH 4 and below). This effect persists at low external pH; however, once the bacteria encounter neutral external pH, both adherence via the adhesin YadA to collagen is significantly increased (Supplementary Fig. 2), and the binding of the cytosolic components is restored (Fig. 2). Bacteria conceivably benefit from this mechanism, which prevents premature effector translocation into any eukaryotic cells in contact in the acidic regions of the gastrointestinal tract, an event that would be energetically expensive and might elicit immune responses. Once the pH-neutral intestine is reached, secretion is restarted within 20-40 min, which is significantly faster than de novo synthesis of injectisomes at this time (Fig. 7). This is in line with earlier observations where assembly of injectisomes by fluorescence microscopy and the resulting secretion was only observed about 60 min after induction of T3SS assembly by temperature shift to 37 °C[17].

We found that similarly to *E. coli*[66,67], *Y. enterocolitica* can partially compensate for acidic external environment, and that at an external pH of 4.0, the cytosolic pH remained at 6.3-6.4 (Fig. 3a). When we used a proton ionophore to create this cytosolic pH at a similar external pH, the cytosolic T3SS components remained bound to the injectisome (Fig. 3b), suggesting that the external pH is not sensed in the bacterial cytosol. A central open question is how exactly the pH change is sensed in the extracytosolic space. We found that overexpression of SctD, which shifts the equilibrium toward association to the injectisome, strongly represses the pH-dependent dissociation of the cytosolic components (Fig. 4). This is not the case for any of the other tested proteins with extracytosolic domains, SctC, SctJ or SctV, pointing out SctD as the prime candidate for conveying the effect of low external pH. The bitopic IM component SctD could react to a drop in external pH with its periplasmic domain and then transmit this signal to the cytosol. Indeed, our data show that the localization of SctD within the membrane clearly differs between external pH of 7 and 4 (Fig. 5). Interestingly, SctD is one of the least conserved genes in the T3SS, especially in comparison to its direct structural neighbors, the highly conserved SctC secretin ring in the OM, as well as SctJ and the export apparatus proteins in the IM[68]. While this low sequence similarity impedes a multi-sequence alignment to identify conserved differences between the T3SS of gastrointestinal and non-gastrointestinal pathogens (Supplementary Table 1), except for the closely related *Y. enterocolitica* and *P. aeruginosa* (Supplementary Fig. 18b), it supports the notion that SctD is involved in species-specific adaptation of the T3SS, such as pH sensing. This hypothesis is further substantiated by the finding that in *P. aeruginosa*, which does not pass the gastrointestinal system during a normal infection, the effect of low pH is significantly restricted, with the majority of EGFP-SctQ foci remaining present at low external pH (Fig. 6). Mutation of candidate amino acids in *Ye*SctD or even complementation of *Y. enterocolitica* ΔSctD with *Pa*SctD did not phenocopy the lack of pH sensitivity, suggesting that a broader

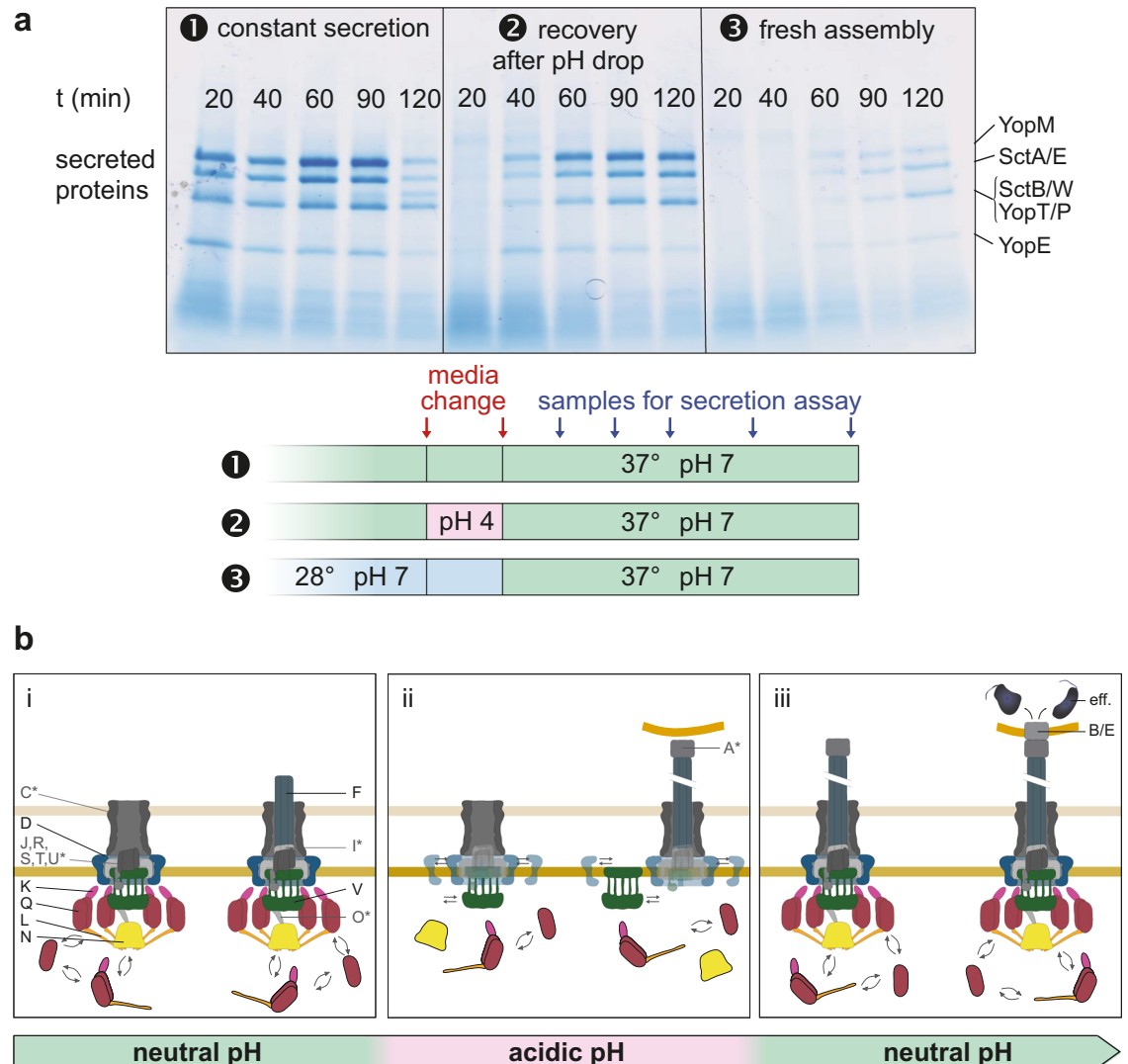

**Fig. 7 Temporary suppression of T3SS activity at low external pH enables a fast reactivation of secretion. a** In vitro secretion assay showing the proteins exported by the T3SS in *Y. enterocolitica* MRS40 at the given time points after the second media change, (1) under constantly secretion-inducing conditions, (2) after a temporary change of the external pH to 4, (3) after incubation at 28 °C, where expression of injectisome components and effectors is downregulated and no injectisomes are assembled. Right side, exported effectors, based on ref. [104,105]. No molecular weight standard was loaded on this gel, please refer to Supplementary Fig. 3a for a direct comparison of molecular weight markers and exported effectors. *n* = 3 independent experiments. **b** Model of the pH-dependent suppression of T3SS activity. From left to right: (i) assembly of the T3SS upon entry into host organisms; cytosolic components bound and exchanging with cytosolic pool. (ii) Prevention of effector translocation upon host cell attachment in low pH environment, because cytosolic components are exclusively cytosolic. (iii) Re-association of cytosolic components to the injectisome and effector translocation upon host cell contact in neutral body parts. Letters represent protein identifiers (common Sct nomenclature); eff., effectors. Proteins whose localization was not specifically tested are indicated by asterisks and gray fill. Straight double arrows indicate diffusion in membrane; curved double arrows indicate the exchange of cytosolic subunits or subcomplexes between the cytosolic and the injectisome-bound state.

reversible conformational change at low pH is responsible for the observed effect. It is currently unclear at which interface(s) this effect takes place; however, the lack of phenotype of over-expression of the other T3SS components with periplasmic domains (SctC, SctJ, and SctV) makes the interface between neighboring SctD molecules the prime candidate for a pH-dependent temporary dissociation.

Both possible types of pH sensing, direct structural change and rearrangement of intermolecular interactions, are used by biological systems: The *E. coli* membrane-integrated transcriptional regulator CadC senses acidic pH through direct protonation of a charged surface patch in its C-terminal periplasmic domain, and transduces the signal to the cytosol by its N-terminal cytosolic domain[69]. Similarly, the *P. syringae* T3SS effector AvrPto

transitions from a largely unfolded state in the mildly acidic bacterial cytosol to a well-defined fold in the neutral host cytosol[70]. Perhaps most strikingly, the highly ordered multimeric structures of bacteriophages undergo large-scale structural changes at low pH, that are reversible[71–73]. Our data show a reversible partial delocalization of SctD at low external pH (Fig. 5). The presence of SctD is required for binding of any cytosolic T3SS component[17,18], most likely through a direct contact of four SctD to one SctK at the cytosolic interface of the IM[15], indicating that this partial displacement of SctD (Fig. 5) is causal for the dissociation of the cytosolic T3SS components.

The dissociation and re-association kinetics of the cytosolic components of the T3SS revealed binding/unbinding half-times of about 1–2 min under secreting conditions (Fig. 2d,

Supplementary Figs. 8–10, Supplementary Movies 4–5). These values are strikingly similar to the exchange rate of SctQ at the injectisome[24], suggesting that at low external pH, primarily the re-association of the cytosolic components is prevented. Notably, recovery of fluorescent foci at pH 7 appears to be faster and more pronounced for the proteins close to SctD (SctK and SctQ) compared to the more distant components (SctL and especially SctN) (Fig. 2); however, both visual and automated spot detection are influenced by the stoichiometry of the respective components, which makes comparisons across different parts of the T3SS difficult.

The mobile fraction of SctD at low external pH diffuses with a coefficient of 0.006–0.048 $\mu m^2/s$, which is similar to other freely diffusing membrane proteins, including the unbound flagellar stator GFP-MotB[74–77]. This finding is compatible with passive diffusion of SctD in the IM under these conditions, which allows a rebinding within few seconds.

SctD is a core component of the injectisome basal body that is reliably copurified in purified needle complexes[78,79] and has an early role in injectisome assembly[68]. While the dissociation of such a central protein upon external cues might be counter-intuitive, dynamic exchange of structural components of large protein complexes and molecular machines has been shown in other cases, in particular for the flagellum (reviewed by Tusk et al.[80]) and the DNA replisome[81], showing that structural and functional roles in a complex and protein exchange are not mutually exclusive.

While the effect of low external pH on the injectisome can be clearly observed in our experiments, it is difficult to estimate its role during infection. Activation of the T3SS by host cell contact can differ from activation by low calcium levels[82] and the exposure of Y. enterocolitica to different pH, as well as the extent of possible host cell contact in the low pH parts of the gastrointestinal system[83] are unclear. In particular, it remains to be determined whether invasin, whose regular downregulation at 37 °C is prevented at acidic pH[84] contributes to host cell binding under these conditions. Nevertheless, the described adaptation of T3SS function at low pH may explain the so-far enigmatic benefit of the dynamic exchange of the cytosolic parts of the injectisome during its function. Notably, the observed dissociation of the cytosolic proteins in response to low external pH may not reveal the complete mechanism for suppression of T3SS activity at this pH. The observation that the dissociation of the cytosolic components occurs at a slightly lower pH (between 4 and 5) than the loss of effector secretion (which occurs between pH 5 and 6), and that the re-initiation of secretion occurs later than the initial recovery of foci indicates that other factors might participate in this phenotype. The slightly delayed activation of secretion after restoration of neutral pH could also be explained by the time required for re-associating SctV to the machinery (the cytosolic components can already bind to the membrane rings in the absence of SctV[17]). Y. enterocolitica may in fact benefit from this lag, as it delays the antiphagocytic effects of the T3SS effectors, which otherwise may hamper the passage of Y. enterocolitica through the M cells that is required to access the lymphoid follicles of the Peyer's patches[85]. As it is currently unclear to which extent Y. enterocolitica and other gastrointestinal pathogens directly contact cells in the low pH parts of the gastrointestinal system, including the stomach, this may be equally or even more beneficial for the bacteria than the direct suppression of secretion at low external pH. Our finding that overexpression of SctD strongly reduces dissociation of the cytosolic T3SS components at low pH, while retaining the secretion ability of Y. enterocolitica might be exploited in future infection experiments to investigate this question.

Like the Y. enterocolitica T3SS, the intracellular Salmonella enterica SPI-2 T3SS is strongly influenced by the external pH;

however, the mechanism described for SPI-2 differs from the one described in this study. Injectisome assembly and secretion of the translocon components in SPI-2 are activated by the low pH of the surrounding vacuole (around pH 5.0)[86,87]. After this step, neutral pH, most likely indicative of a successfully established connection to the neutral host cytosol, leads to the disassembly of a gatekeeper complex, which in turn licenses the translocation of effectors[87]. Strikingly, a single amino acid exchange in the export apparatus protein SctV, V632D, governs this pH-dependent effect[88]. Both the sensory and the functional connection between the pH and T3SS activity differ between Salmonella SPI-2 and Y. enterocolitica, which highlights the high degree of functional adaptability of the T3SS.

In this study, we found that dynamic exchange of cytosolic components of the T3SS enables to suppress the activation of the T3SS at low external pH in Y. enterocolitica. Strikingly, this effect was also observed in S. flexneri, indicating a conservation in gastrointestinal pathogens, but not in P. aeruginosa, which does not pass low pH regions during normal infections. Our data highlight that not only the effector repertoire, but also dynamics and function of T3SS are adapted to the specific environment of the respective bacteria. This newly described pathway is a striking showcase for the variety and sophistication of mechanisms that allow bacteria to use different aspects of T3SS assembly and function, including protein dynamics, to tailor the activity of this essential virulence mechanism to their specific needs during infection.

## Methods

**Number of replicates and statistical analysis**. The number of replicates is indicated with the data for all quantitative analyses. Where not indicated in cases where representative images are shown, experiments were performed for at least three independent biological replicates. A two-tailed homoscedastic t-test was used for statistical analysis. Unless specified differently, error bars display the standard deviation, except for discrete measurements of low numbers, such as the number of fluorescent foci per bacterium, where the standard error of the mean is displayed.

**Bacterial strain generation and genetic constructs**. A list of strains and plasmids used in this study can be found in Supplementary Table 2. The Y. enterocolitica strains used in this study are based on the Y. enterocolitica wild-type strain MRS40[89], and the strain IML421asd (ΔHOPEMTasd), in which all major virulence effector proteins (YopH,O,P,E,M,T) are deleted[90]. Furthermore, this strain harbors a deletion of the aspartate-beta-semialdehyde dehydrogenase gene which render the strain auxotrophic for diaminopimelic acid (DAP).

Unless specifically mentioned otherwise, all Y. enterocolitica fusion proteins used in this study are expressed as endogenous fusions from their native location on the pYV virulence plasmid; the genes were introduced by allelic exchange[91] and replace the respective wild-type genes.

For cloning and conjugation the E. coli strains Top10 and SM10 λpir were used, respectively. All constructs were confirmed by sequencing (Eurofins Scientific and Microsynth Seqlab). Constructs with single amino acid substitutions in SctD or SctF and other expression constructs were created by overlapping PCR using Phusion polymerase (New England Biolabs), and expressed from an arabinose controlled expression vector (pBAD).

**Bacterial cultivation, in vitro secretion assays and fluorescence microscopy**. Y. enterocolitica day cultures were inoculated from stationary overnight cultures to an $OD_{600}$ of 0.15 and 0.12 for secreting and non-secreting conditions, respectively, in BHI medium (Supplementary Table 3) supplemented with nalidixic acid (35 mg/ml), diaminopimelic acid (80 mg/ml), glycerol (0.4%) and $MgCl_2$ (20 mM). Where required, ampicillin was added (0.2 mg/ml) to select for pBAD-based plasmids. For secreting conditions, cultures were additionally supplemented with 5 mM EGTA; for non-secreting conditions, cultures were additionally supplemented with 5 mM $CaCl_2$ and filtered through a 0.45 μm filter. Cultures were incubated at 28 °C for 90 min. At this time point, expression of the yop regulon was induced by a rapid temperature shift to 37 °C in a water bath and expression of proteins in trans was induced by the addition of 0.2–1.0% arabinose, as indicated.

For protein secretion or total cell analysis, bacteria were incubated at 37 °C between 1 and 3 h, as indicated. 2 ml of the culture were collected at 21,000 × g for 10 min. The supernatant was separated from the total cell fraction and precipitated with trichloroacetic acid (TCA) for 1–8 h at 4 °C; proteins were collected by centrifugation for 15–20 min at 21,000 × g and 4 °C, washed once with ice cold acetone and then resuspended and normalized in SDS-PAGE loading buffer. Total

cell samples were normalized to $2.5 \times 10^8$ bacteria and supernatant samples to the equivalent of $3 \times 10^8$ bacteria per SDS-PAGE gel lane. Samples were incubated for 5 min at 99 °C, separated on 12–20% SDS-PAGE gels, using BlueClassic Prestained Marker (Jena Biosciences) as size standard, and stained with InstantBlue (Expedeon). Immunoblots were carried out using a primary rabbit antibody against Y. enterocolitica SctD (MIPA232, 1:1000) in combination with a secondary antibody goat anti-rabbit, conjugated to horseradish peroxidase (Sigma A8275, 1:5000), and visualized using ECL chemiluminescent substrate (Pierce) on a LAS-4000 Luminescence Image Analyzer. Uncropped and unprocessed scans of all gels and blots are available in the Source data file.

For fluorescence microscopy, bacteria were treated as described above. After 2–3 h at 37 °C, 400 μl of bacterial culture were collected by centrifugation ($2400 \times g$, 2 min) and resuspended in 200 μl microscopy minimal medium (Supplementary Table 3). Cells were spotted on 1.5% low melting agarose pads (Sigma) in microscopy minimal medium in glass depression slides. Where required, 80 mg/ml DAP for IML421asd (ΔHOPEMTasd)-based strains, L-arabinose for induction of in trans expression, 5 mM Ca$^{2+}$ for non-secreting conditions, or 5 mM EGTA for secreting conditions were added.

For the fluorescence microscopy experiment comparing different bacterial species, Y. enterocolitica were treated as above, except for the final resuspension for microscopy, which was done in BHI supplemented with 50 mM glycine, 50 mM MES, 50 mM HEPES, 80 mg/ml DAP, and 5 mM EGTA and adjusted to the indicated pH for all species. P. aeruginosa overnight cultures were grown at 28 °C. BHI medium (Supplementary Table 3) was supplemented with 20 mM MgCl$_2$, 200 mM NaCl, and 5 mM EGTA and inoculated to an OD$_{600}$ of 0.15 and incubated for 2 h at 37 °C. This culture was then used to inoculate a fresh culture containing the same supplements to an OD$_{600}$ of 0.15. After 2 h, bacteria were collected and prepared for microscopy as described above for Y. enterocolitica. S. flexneri overnight cultures where inoculated from a fresh LB-agar plate or glycerol stocks into BHI broth supplemented with ampicillin (0.1 mg/ml) and kanamycin (0.05 mg/ml). On the next day, fresh BHI cultures were inoculated to an OD of 0.12 and grown to an OD of around 1 at 37 °C[60]. At this point, bacteria were collected and prepared for microscopy as described above for Y. enterocolitica. For the quantification of the fraction of T3SS-positive bacteria, 530–1143 bacteria in 21–25 fields of view from three independent experiments per strain and condition were classified independently by two blinded researchers and the average numbers were used for the graphs.

**Wide field fluorescence microscopy**. Microscopy was performed on a Deltavision Elite Optical Sectioning Microscope (Applied Precision), equipped with a UApo N 100×/1.49 oil TIRF UIS2 objective (Olympus) and a UPlanSApo 100×/1.40 oil objective (Olympus), using an Evolve EMCCD Camera (Photometrics). Samples were illuminated for 0.1 s with a 488 nm laser with a TIRF depth setting of 3440 (TIRF) or for 0.2 s with green LED light (standard oil objective). Excitation and emission filter sets used were 475/28 and 525/48 nm (green), 575/25 and 625/45 nm (red), or 632/22 and 679/34 nm (Cy5). The micrographs where deconvolved using softWoRx 7.0.0 (standard "conservative" settings). Images were then further processed with Fiji (ImageJ 1.51f/1.52i/1.52n). Where necessary, drift correction was performed with the StackReg Plugin (ref. [92], distribution dated July 7, 2011). For figures, representative fields of view were selected, and brightness and contrast of the micrographs was adjusted identically within compared image sets.

Sample sizes and number of replicates for wide field microscopy and all other experiments were determined prior to the experiments, based on experimental feasibility and their potential to draw clear conclusions. These numbers were not changed based on the experimental outcome.

**Flow cell based TIRF microscopy**. A microscopy flow cell was manufactured based on ref. [93]. Formation of injectisomes was induced under non-secreting conditions. After formation of the injectisomes (150–180 min at 37 °C), bacteria were collected and resuspended in ~0.5 volumes of microscopy medium (Supplementary Table 3) supplemented with 80 mg/ml DAP and 5 mM EGTA. The flow cell was pre-incubated with microscopy medium and a bottom coverslip (25 mm No. 1.5; VWR) was attached to the cell. Bacteria were spotted on the coverslip and incubated for 1 min to allow attachment to the glass surface. The coverslip was covered with 60 μl of the minimal medium and sealed from the top with an additional coverslip. The flow cell was mounted on the microscopy stage and gravity-driven buffer flow in the chamber was induced using a 1 ml syringe. To exchange the buffer in the flow cell, the tube was quickly relocated to a new reservoir. Complete buffer exchange in the flow cell was determined to take 39 ± 5 s ($n = 7$).

**Detection and quantification of fluorescent foci**. Detection and quantification of foci was performed by first segmenting the images using the software BiofilmQ[94]. After segmentation, fluorescent foci were detected using a customized script within BiofilmQ v0.1.3. Foci were identified by finding local maxima within the bacterial cells prior to filtering by their quality, which was defined to be their brightness above the cell´s background intensity given by subtracting a smoothed image, divided by the overall mean fluorescence intensity of bacterial cells.

**Quantification of fluorescent foci upon change of external pH**. Bacteria were prepared for microscopy under secreting conditions as described above. After 2.5 h at 37 °C, bacteria were loaded into the flow cell and buffer exchange from pH 7 to pH 4 and vice versa was performed. Cells were tracked by TIRF microscopy. Micrographs were acquired every 10 s and quantification of foci was performed with BiofilmQ as described above.

**Maleimide based needle staining**. To visualize the injectisome needles, expression of SctF$_{S5C}$[46] was induced from plasmid. Protein expression was induced with 0.4-1.0% L-arabinose at the temperature shift. After 2–3 h, bacteria were collected and resuspended in 0.2 volumes of microscopy medium supplemented with 5 μM of CF 488A/568/633 maleimide dye (Sigma-Aldrich, USA) for 5 min in 100 μl of minimal medium at 37 °C. After staining, bacteria were washed once with 500–1500 μl of minimal medium and spotted on 1.5% agarose pads in the same medium or analyzed in a flow cell as described above.

**Halo staining with Janelia fluorescent dyes**. To stain and visualize a defined pool of T3SS proteins, Halo-labeled proteins were visualized using Janelia Fluor 549-NHS ester or Janelia Fluor 646-NHS ester. 500 μl of bacterial culture were collected by centrifugation ($2400 \times g$, 2 min) and resuspended in 100 μl microscopy medium. Bacteria were stained with 0.2 μM Janelia Fluor JF 646/JF 549 dyes at 37 °C for 30 min in a tabletop shaker. Afterward, bacteria were washed twice in 500 μl minimal medium and then visualized in a flow cell or on 1.5% agarose pads in the same medium.

**Equilibration of cytosolic and external pH**. 2,4-dinitrophenol (DNP) was diluted in methanol and used at a final concentration of 2 mM in microscopy minimal medium. Cells were grown as described above, and resuspended in DNP containing medium immediately before being spotted on 1.5% agarose pads in the same medium containing DNP.

**Bacterial survival test**. Wild-type Y. enterocolitica MRS40 were inoculated to an OD$_{600}$ of 0.12 in non-secreting conditions and incubated for 1.5 h at 28 °C to reach exponential growth phase. Cells were collected by centrifugation at $2400–3000 \times g$ for 2–5 min. The supernatant was discarded and the collected bacteria were resuspended in a range of pH-adjusted media buffered with 50 mM glycine, 50 mM HEPES and 50 mM MES. Cultures were then placed in a 37 °C shaking water bath and incubated at the indicated pH for 30 min. A dilution series in 1:10 steps was performed. 4 μl of the bacterial dilution were plated on a LB agar plate supplemented with nalidixic acid (35 mg/ml). Droplets were dried for 15 min before incubation of the plate overnight at 28 °C.

**Growth curve assays**. Wild-type Y. enterocolitica MRS40 were inoculated to an OD$_{600}$ of 0.12 in secreting conditions and incubated for 90 min at 28 °C. At this point, cells were collected by centrifugation ($2400 \times g$, 2 min), and resuspended in 37 °C fresh pre-warmed secreting medium buffered with 50 mM glycine, 50 mM HEPES and adjusted to the indicated pH. After 90 min, one batch of cells incubated at pH 4 were collected again and resuspended in fresh buffered secretion medium at pH 7. All strains were incubated for further 90 min. During the experiment, the OD$_{600}$ was measured every 30 min in technical duplicates, which were averaged; the figure displays the single data points for three independent biological replicates.

**pHluorin purification and calibration**. The protocol described by Nakamura et al.[95] was adapted for bench top purification. Expression of ratiometric GST-pHluorin$_{M153R}$ in E. coli DH5αZI was induced in a 500 ml culture with 1 mM IPTG, followed by 4 h incubation at 28 °C. Cells were harvested and stored for further processing at −80 °C. Next, cells were thawed on ice, resuspended in lysis buffer and disrupted by two passages through a French Press. The lysate was centrifuged ($75,600 \times g$, 60 min, 4 °C) and the supernatant was gently mixed with the previously equilibrated glutathione agarose at 4 °C for 1.5 h in a small spinning wheel. The agarose was collected by centrifugation ($500 \times g$, 5 min) and washed with PBS three times. A thrombin digest was performed in 2 ml PBS (16 units) while incubating at room temperature for 60 min on a roll mill. Agarose was removed by centrifugation ($500 \times g$, 5 min) and the protein was stored at −80 °C for further use.

The calibration was performed on a Deltavision Elite microscope. 5 μl of purified pHluorin protein were spotted on a KOH-cleaned microscopy slide in an enclosed compartment (Thermo Fisher Scientific GeneFrame, AB-0577) and incubated for 5 min to ensure attachment to the glass surface. Then, 10 μl of pH-adjusted PBS buffered with 200 mM glycine, 200 mM HEPES, and 200 mM MES were added. Ratiometric pHluorin fluorescence was determined using the DAPI excitation filter (390/18 nm) or GFP excitation filter (475/28 nm) at 32% illumination intensity, in combination with a GFP emission filter set (525/48 nm) with 0.3 s exposure time. The overall fluorescence of the images before deconvolution was determined and corrected for background fluorescence levels, which were measured with an empty slide filled with PBS. The ratio of fluorescence intensity using DAPI/Green vs. Green/Green excitation/emission filters was determined for a pH range from pH 8–pH 5 in 0.5 pH steps.

## Single-particle tracking photoactivated localization microscopy (sptPALM).

Bacteria were cultivated under non-secreting conditions as described above. After 2.5 h of incubation at 37 °C, the media was changed to pre-warmed (37 °C) minimal microscopy medium containing the same supplements as the BHI before. Cells were incubated for 30 min and then washed four times in 5 volumes of pre-warmed (37 °C) EZ medium (Supplementary Table 3) supplemented with DAP and 5 mM CaCl$_2$. Bacteria were concentrated 2× after the last wash and spotted on a 1.5% agarose pad in EZ medium supplemented with DAP and 5 mM CaCl$_2$ on a KOH-cleaned microscopy slide in an enclosed compartment (Thermo Fisher Scientific GeneFrame, AB-0577). Imaging was performed on a custom build setup based on an automated Nikon Ti Eclipse microscope equipped with appropriate dichroic and filters (ET DAPI/FITC/Cy3 dichroic, ZT405/488/561rpc rejection filter, ET610/75 bandpass, Chroma), and a CFI Apo TIRF 100x oil objective (NA 1.49, Nikon). All lasers (405 nm OBIS, 561 nm OBIS; all Coherent Inc. USA) were modulated via an acousto-optical tunable filter (AOTF) (Gooch and Housego, USA). Fluorescence was detected by an EMCCD camera (iXON Ultra 888, Andor, UK) in frame transfer mode and read-out parameter settings of EM-gain 300, pre-amp gain 2 and 30 MHz read-out speed. The z-focus was controlled using a commercial perfect focus system (Nikon). Acquisitions were controlled by Micro-Manager v1.4.23[96]. Live cell sptPALM experiments were performed on a customized heating stage at 25 °C. Live *Y. enterocolitica* PAmCherry-SctD cells were imaged in HILO illumination mode[97]. Applied laser intensities measured after objective were 35 W/cm$^2$ (405 nm) and 800 W/cm$^2$ (561 nm). Prior to recording each new region of interest (ROI), a pre-bleaching step of 561 nm illumination was applied for 30 s to reduce autofluorescence. Videos were then recorded for 2000 frames pulsing the 405 nm laser every 20th imaging frame at 5 Hz with an exposure time of 200 ms per frame. After sptPALM imaging, a bright light snapshot of all illuminated regions was recorded to obtain the bacterial cell shapes.

Single molecule localizations were obtained using rapidSTORM 3.3.1[98] and single cells were manually segmented in Fiji (ImageJ 1.51f/1.52i/1.52n)[99]. sptPALM data were tracked, visualized and filtered using the swift tracking software 0.3.1. Trajectories were allowed to have a maximum of 5 frames of gap time (e.g., caused by fluorophore blinking). Trajectories were assigned to their diffusive states (static and mobile) on the basis of the experimental localization precision of about 25 nm (determined by the NeNA method[100]). For trajectories with more than 6 one-frame jumps, the mean jump distance (MJD) was calculated (jumps spanning several frames due to dark times were not used in MJD calculations). The obtained MJDs were weighted by the number of jumps and displayed in a histogram using OriginPro 9.4 (Origin LAB Corporation, USA). To estimate the diffusion coefficient D from the MJDs (M) at the frame rate $\tau = 200$ ms and the localization precision $\sigma_m = 25$ nm, the equation $D = (M^2\pi^{-1} - \sigma_m^2)\tau^{-1}$ was used.

## *Y. enterocolitica* YadA and cellular adhesion assays.

To measure YadA adhesion to collagen at different pH values, an ELISA-like binding assay was performed[101–103]. Clear 96 well plates (Sarstedt, Ref. 82.1581) were coated with 100 µl of calf collagen type I (10 µg/ml, ThermoFisher A1064401) in H$_2$O with 0.01 M acetic acid for 1 h at room temperature (RT). The supernatant was discarded and the wells were blocked with 1% BSA in PBS and afterward washed three times with 0.1% BSA in PBS. Afterward, 100 µL of purified YadA head domains with a C-terminal His$_6$-tag at a concentration of 10 µg/mL were incubated in the wells. In order to test binding at different pH, the YadA heads were diluted in acetic acid/sodium acetate at pH 4.0 or pH 5.0 and in PBS for higher pH values. Binding was allowed for 1 h at RT. Afterward, the plate was emptied and washed three times with 0.1% BSA in PBS. The wells were blocked with 1.0% BSA in PBS for 1 h at RT. For detection, Ni-HRP conjugates (HisProbe-HRP, Thermo Fisher Scientific, Ref. 15165) were diluted in 0.1% BSA in PBS and 100 µl per well were incubated for 1 h at RT. The Ni-HRP conjugate solution was discarded and the wells washed three times with PBS. Detection was performed using ABTS substrate (Thermo Scientific, Ref. 34026). Development was allowed for 30 min. Absorption was measured at 405 nm on a plate reader (Biotek Synergy H1).

Glass binding assays for live *Y. enterocolitica* were performed in a flow cell. Bacteria were treated as mentioned above, resuspended in microscopy medium at the indicated pH value, and added to the flow cell as described above. Bacteria were tracked visually for 10 min afterward.

## Reporting summary.

Further information on research design is available in the Nature Research Reporting Summary linked to this article.

## Data availability

All relevant data are included in the paper and/or its Supplementary information files. Source data are provided with this paper.

## Code availability

Version 0.3.1, used in this manuscript, and all subsequent versions of the swift software, as well as documentation and test data sets, can be obtained on the swift beta-testing repository (http://bit.ly/swifttracking) and upon request to the authors.

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

## Acknowledgements
We thank Nicholas Dickenson (Utah State University, Logan, USA) for the kind sharing of *Shigella flexneri* GFP-Spa47, Sophie Bleves (Aix-Marseille University, FR) for the *Pseudomonas aeruginosa* PAO1 strain, Marc Erhardt (Humboldt University Berlin, Germany), Tohru Minamino (Osaka University, Japan), and Gero Miesenböck (University of Oxford, UK) for the pHluorin$_{M153R}$ DNA, Luke Lavis (Janelia Research Campus, USA) for the kind donation of Janelia Fluor dyes, and Malte Buchholz (University of Marburg, Germany) for helpful discussions about gastrointestinal infection pathways. We are also grateful to Carlos Helbig and Bartosz Turkowyd (Max Planck Institute Marburg, Germany and Carnegie Mellon University, USA) for analysis scripts. This work was supported by the Max Planck Society. It was also funded by the Horizon 2020 Innovative Training Network "ViBrANT" (to D. Linke).

## Author contributions
S.W. performed the majority of the experiments and data analysis and participated in study design and writing the manuscript. A.B. performed and analyzed the sptPALM experiment with support from S.W. H.J. provided coding for the automated spot detection. L.S., D. Lampaki and E.E. assisted in experiments and strain generation. I.M. performed the YadA binding experiments and participated in proofreading the manuscript. D. Linke, K.D., and U.E. provided supervision and participated in discussions and proofreading. A.D. conceived and designed the study, performed experiments and data analysis and wrote the manuscript.

## Funding

## Competing interests
The authors declare no competing interests.
