## [Peer Review File · Nature Communications]

REVIEWER COMMENTS

Reviewer #1 (Remarks to the Author):

This manuscript from the Diepold group describes reorganization of the *Yersinia* T3SS injectisome at low external pH as a mechanism for preventing premature effector secretion while in the stomach. The authors are experienced at exploring the dynamics of the injectisome cytosolic components during T3SS activation in *Yersinia* and they have expanded that work here.

The proposed concept is certainly novel and is largely supported by the data they present, however, there are some rather important concerns that need to be considered by the authors and addressed the manuscript. The disassembly of the injectisome cytosolic components at low environmental pH seems to be clearly presented, but the role of the injectisome basal body in this process may need to be more critically evaluated. Below are some issues that the authors should address to make the work more complete.

MAJOR CONCERNS

In Suppl Fig 9, there is a clear co-localization of SctF with SctL under the initial pH 7 conditions, however, this colocalization at pH 4 and upon return to pH 7 appear to be very similar. In some other places the data seem to be open to multiple interpretations. For example, in Fig 2, the return to the original pH 7 state for SctL and SctN doesn't seem to be so clear cut though it does appear so for SctQ and SctK. These are challenging experiments to perform, so rigorous critical evaluation of the data is essential. The authors should discuss the potential difficulties in data analysis and interpretation.

In Suppl Fig 12, the data for *Shigella* are not quite as convincing as the authors state and the question that arises is whether or not there is a possibility that *Shigella* can actually invade the epithelium of the stomach. This and the fact that *Pseudomonas aeruginosa* actually appears to form more and better defined punctate foci warrants mentioning.

As the authors begin to propose how such events might be triggered, they home in on SctD and SctV. Several recent papers propose that SctV undergoes rearrangement in the absence of the injectisome cytosolic components (perhaps as a checkpoint to prevent premature effector secretion), however, the authors spend very little time discussing SctV and focus largely on SctD. In the absence of defining the residues of SctD that might be involved in disassembly of the injectisome cytosolic components, the authors should more critically consider the role of SctV in this process.

For the SctD experiments, it is not clear that the EGFP-SctD fusion was tested to see if it retained its T3SS secretory activity or allowed assembly of the injectisome cytosolic components. This needs to be done or made more apparent that it was done. The EGFP probe appears to be at the SctD N terminus which would place it on the cytoplasmic domain of the protein which is the domain through which it interacts with SctK. This domain is small (~13kDa) and the mass of EGFP is on the order of 33kDa, which could sterically influence its interaction with SctK. Likewise, it's unclear how this added density would influence the packing of the inner membrane ring or even the interaction between SctD and SctJ or SctC. Any compromise of the IM ring stability should be discussed.

With respect to the model proposed for SctD dissociation from the injectisome, the authors should consider the following: SctD is a rather stable component of the injectisome that, unlike the injectisome cytosolic components, remains associated with the complex even after somewhat harsh purification processes. Likewise, assembly of the IM ring has been proposed by multiple groups and in reviews as being an early step in overall injectisome assembly. For it to later disassociate from the injectisome seems somewhat counterintuitive. The authors should reconcile

this assumption with their model.

Related to the above comment, it appears in the proposed model that dispersal of SctD in the IM occurs via passive diffusion. Is this the case or is it a directed process as might be suggested by the ability to rapidly reassemble back into an active complex when the pH is neutralized? Furthermore, how might membrane fluidics play a role in dispersion and reassembly?

With respect to the mutational analysis of SctD, removal of key charged amino acids had no effect on SctQ dispersal, but the authors do comment that *Pseudomonas* and *Yersinia* SctD are relatively well conserved (Suppl Fig 12B). Thus, could the authors substitute the entire protein or the entire periplasmic domain to test their model? In doing so, the authors may find out that either the trigger is not SctD, but maybe SctJ (or that SctV may have a major role here). Likewise, this might help in determining whether the fused EGFP is partially destabilizing to the IM ring.

MINOR CONCERNS

The writing could use a little smoothing out (paragraph formats, etc.) and a bit more time spent discussing the results would be helpful.

Question: Is there evidence that YadA contributes to adherence to the mucosal surface in the stomach? In other words, is T3SS secretion really a concern physiologically in the stomach?

Related to the comment above, upon ingestion the temperature is elevated to 37C at which time the T3SS begins to be expressed. Fig 6 suggests that full expression of the T3SS injectisome takes about 2 hours, but Fig 1B suggests that the bacteria pass through the gut in as little as 30 minutes and certainly by 4 hours. Based on this, do the authors consider accidental activation of the T3SS while in the stomach to be a problem the pathogen must overcome?

Reviewer #2 (Remarks to the Author):

In this manuscript, the authors report on the finding that gastrointestinal pathogens with T3SS utilize pH change as a means of regulating the timing of effector protein secretion into a host cell. They demonstrate that *Yersinia enterocolitica* cultures remain viable after incubation at a pH equivalent to that of passage through the stomach and that exposure to this acidic environment inhibits the secretion of effector proteins without affecting the localization of T3SS needle components to specific foci at the bacterial membrane. The authors also showed that while secreted effector proteins do not colocalize with the needle complex at low pH, the secretion phenotype is restored upon reintroduction to physiological pH. They hypothesized that the periplasmic T3SS protein SctD was involved in external pH sensing and regulating effector secretion and found that whereas SctD did not colocalize with needle foci at low pH, changes in external pH did not alter the protonation state of SctD. Among the species they tested, this behavior was only observed in T3SS pathogens of the GI tract.

Overall, the data presented are interesting, the quality of the data is outstanding, and the data support the authors' conclusions. I have a few major concerns that should be addressed regarding controls that are needed to support their key findings.

Major concerns:

1. A critical missing control is whether the fluorescent signal per se is decreased by low pH. Indeed, in a paper from GH Patterson et al. (*Biophys J* 1997), the fluorescence of EGFP decreased more than 80-fold when the fluorophore was shifted from pH 7.0 to pH 4.0. The authors need to perform control experiments to test whether this is occurring here for each of the fluorophores they use (they could use a cytoplasmic EGFP and mCherry and quantify their signal intensities under

conditions in which the cytoplasm is equilibrated to this range of pH's. Although the increase in cytosolic fluorescence at pH 4 reported by the authors in Fig. 2C would argue against pH-dependent decreases in fluorescence per se, the experiment shown in Fig. 2C is not sufficient as this control, as the two conditions being compared also differ in other important respects.

2. Since *Yersinia pestis* bypasses the intestine, being instead introduced into the host by injection from fleas into the bloodstream, the author's model would predict that the *Y. pestis* SctN and SctQ homologs would differ from those of *Y. enterocolitica* and that localization in *Y. pestis* would be independent of pH. This seems likely to be a better comparator than *P. aeruginosa*. The authors should examine this and potentially residues in the SctD homologs that differ between the two species and which might be protonated at pH 4 but not at pH 7. (For experiments, to avoid any biosafety issues, the vaccine strain of *Y. pestis* could be used.)

3. Statistics: for each set of experiments, the authors should indicate the number of independent experiments (e.g., Fig. 4B) and provide statistical significance appropriate based on independent replicates. In some cases, it is difficult to tell whether the data presented represent biological or technical replicates (e.g., Fig. 5).

Minor comments:

1. Line 16: Please clarify "proliferate in contact to..."

2. Line 80: Figure 1B shows that *Yersinia* persists in the stomach for a minimum of 30 minutes. Explain why 15 minutes was used as the experimental condition.

3. Lines 82-83: Optical density does not equate to viability as cells could be dead but still intact and thereby contribute to the optical density of a culture. This suggests the possibility that by exposing the cultures to a low pH, the authors were selecting for a subpopulation of bacteria. Moreover, in Suppl. Fig. 1, there appears to be a fitness decrease, as half of the cells appear to die when exposed to a low pH, and these cells are slower to recover when returned to physiological pH. The authors should be less dogmatic about their conclusions that low pH had no fitness selection effect.

4. Please include statistical tests in Figure 1D

5. Line 88: What general surfaces? Abiotic surfaces?

6. Please add error bars on Suppl. Fig. 7.

7. Figure 3A, right panel: error bar for 10 minutes at pH4 is not consistent with the rest.

8. Figure 4B, pH7: it is unclear whether the dynamics of SctD are T3SS specific or typical of all inner membrane proteins. A useful control might be a known static membrane protein.

9. Lines 276-277: Please clarify whether *Shigella* and *Pseudomonas* are expressing recombinant *Yersinia* SctQ or the *Shigella* and *Pseudomonas* homologs to SctQ.

10. Figure 5: Please include error bars and statistics.

11. Line 295: "Showever"

12. Figure 6A: Clarify whether foci are present and effectors and needle components are being expressed at 28C.

13. Line 351 and Figure 4: The authors show that SctD dissociates from foci at low external pH but that this dissociation was not due to differential protonation states of SctD. Given that there are other periplasmic components of the T3SS injectisome, it is also possible that SctD does not directly sense external pH and instead, another periplasmic protein undergoes conformational changes as a direct result of the drop in external pH resulting in dissociation of SctD from the complex. These possibilities (and others) should be discussed.

14. Figure 12B: The authors showed that *Pseudomonas* effector secretion via T3SS does not respond to low external pH as *Yersinia* does, but that *Yersinia* and *Shigella* both respond similarly to low external pH. It would be informative to compare the sequences of periplasmic T3SS proteins between *Yersinia* and *Shigella* and use the similarities as a cross reference against the differences found between *Yersinia* and *Pseudomonas*.

15. Lines 391-392: Please clarify "dissociation of the cytosolic components occurs at a slightly lower pH than the loss of effector secretion".

16. Line 398: The authors carried out secretion experiments in BHI, which is dissimilar to the intracellular environment of an M cell, therefore cannot conclude that we will see the same

secretion delay in an M cell as observed in BHI. Would be careful not to overstate the conclusions here.

17. Lines 406-407: Clarify whether the amino acid in SctV is susceptible to protonation changes.

18. Line 497: "we fluorescent foci"

Reviewer #3 (Remarks to the Author):

Wimmi et al. present a thorough study on the pH-dependent machinery for bacterial injection of molecular toxins into host cells. The study reveals a molecular mechanism for the dissociation of the cytosolic T3SS components in response to extracellular pH variations and thus to environmental control of the bacterial injection of molecular toxins. The study resolves the contribution of various T3SS protein components to the pH-dependent disassembly of the T3SS complex. Fluorescence microscopy was used to monitor the dissociation of the T3SS complex. Single-molecule tracking was employed to observe changes in the molecular mobility of SctD and reveal its role as modulator for pH sensing. Observations for various environmental conditions and bacterial pathogens confirmed that the observed mechanism is of general relevance.

In my opinion, this is an excellent manuscript. The study is well designed and of superb quality with regard to experimental work, data analysis and data interpretation. The fluorescence microscopy data is of high quality. Quantitative analysis is carried out in great detail and all statistical information is provided. The analysis allows for great confidence on the correct interpretation of the presented data. Without any hesitation I like to recommend publication of the manuscript in Nature Communications.

Reviewer #4 (Remarks to the Author):

The manuscript by Wimmi et al describes a phenomenon whereby several cytosolic injectisome components appear to dissociate from the inner membrane component SctD/YscD upon shift to low pH. This dissociation leads to a dramatic reduction in secretion of T3SS components and is reversible which allows the bacteria to rapidly commence secretion at high pH. The authors propose that this observation provides a means of preventing secretion of Yops in the stomach after oral ingestion of *Yersinia* (and *Shigella*), but rapidly ramping up secretion as the bacteria experience a rise in pH towards the distal end of the small bowel. The idea is intriguing and the authors data with various forms of microscopy using fluorescent probes support their model. Much of the data is rigorously controlled with ample data sets taken from 2 or more experiments, although in some panels it is unclear how many experiments were performed and how many cells evaluated for the conclusions drawn (for instance, 2D, 2E, 3B, 4C). The paper is generally well written for the broad readership of Nature Communication and the findings are of interest to the audience. Several weaknesses are noted in points below:

1. The mechanism that controls the low pH response is not identified in the manuscript. The authors propose it is SctD since SctD also disperses upon low pH, but the evidence provided that shows SctD senses pH is missing. The authors show that in *Yersinia*, SctD become diffuse in membranes in low pH, while in *Pseudomonas*, SctD's 'diffusion' in membranes is the same regardless of pH (and intermediate between that found in *Yersinia* at pH7 versus pH 4), but this does not pin down *Yersinia* SctD as the molecule that senses low pH.

2. The authors make 4 amino acid substitutions of SctD in histidine, but these do not have a phenotype, so this also does not pin down the component sensing low pH. However, there are about 18 His, Glu and Asp in the periplasmic portion of SctD that are not found in *Pseudomonas*. Swapping out groups of these to support or eliminate their hypothesis is possible (or substituting the periplasmic portion of the *Pseudomonas* SctD for the *Yersinia* and determining if secretion is

occurring, if these components are interchangeable as some components are). While the homology is very weak, can the authors try to suss out whether a subset of the potential H, D and Es are structurally conserved between Shigella and Yersinia but not Pseudomonas and substitute these?

3. Some of their reassembly data buried in the supplemental materials (figures 8-9, especially Suppl Fig 9) were potentially quite intriguing and could have been highlighted more if shown in a rigorous manner. Information about the number of times these images were evaluated is missing, so it is unclear how strong these data and the points the authors are trying to make in the absence of rigorous evaluation. For instance, in the image in Suppl Fig 9 at pH4, mCherry-SctL does seem to colocalize even at pH4 and not much more is associated after re-shift to pH 7.
4. All of the dissociation/diffusion is measured by various fluorescence techniques. An independent method for instance, crosslinking under both conditions or IPs, would strengthen these findings.
5. Virtually all of their findings show that at low pH, T3SS components disaggregate with the exception of the SctF, but that molecule is external and we do not know whether the needles are still largely intact or if in fact they have mostly fallen apart and the probe is detecting SctF associated with SctC. A control is needed showing that low pH is not causing significant disruption of most membrane and membrane associated proteins in Yersinia. The best control is the Pseudomonas in Suppl Fig 12, but these bacteria are grown in different media.
6. The authors spend some time justifying their experiments by proposing that this mechanism is needed to prevent Yersinia from injecting Yops into cells in the stomach. But do cells lining the stomach express beta-1 integrins - the integrins that Yersinia primarily uses to attach to cells? Most cells in the intestine do not express beta-1 integrins - and it has been shown in mice at least that Yersinia bind to a very small population of cells in the intestine. Is there any evidence that Yersinia is more likely to invade stomach lining in mice treated with a proton pump inhibitors? (The data showing binding is in Fig 1D is not directly showing that attachment to cells occurs at pH 4.) Certainly, a more nuanced discussion of this rationale is required.

Minor points:

1. Line 22 - more up to date references/reviews on Yersinia effector functions would be good - one of these, although wonderful is 22 years old and the other 13 years old. The field has made some advances since then.
2. Line 62 - more up to date references on translocation.
3. Lines 137-139: Suppl Fig 6 seems to show that the dissociation kinetics are similar between secreting and non-secreting conditions for the proteins shown, not that it is reversible as described in lines 133-135 and shown in Supple Fig 5
4. Line 295 - change Showever to However
5. Suppl Fig 6. The legend states that rows 1 and 3 are secreting and rows 2 and 4 are non-secreting, but the figure shows otherwise.
6. I find giving all these components from different bacteria, the exact same name from when they function differently and often have poor sequence homology to each other, as the authors describe here functionally with the Pseudomonas system versus the Yersinia/Shigella system (or the Shigella versus Yersinia/Pseudomonas genes), problematic. While it does simplify remembering them all, it adds confusion when writing. Perhaps calling them SctDy versus SctDp for Yersinia version versus Pseudomonas would help resolve the conundrum.

REVIEWER COMMENTS

Reviewer #1 (Remarks to the Author):

This manuscript from the Diepold group describes reorganization of the *Yersinia* T3SS injectisome at low external pH as a mechanism for preventing premature effector secretion while in the stomach. The authors are experienced at exploring the dynamics of the injectisome cytosolic components during T3SS activation in *Yersinia* and they have expanded that work here.

The proposed concept is certainly novel and is largely supported by the data they present, however, there are some rather important concerns that need to be considered by the authors and addressed in the manuscript. The disassembly of the injectisome cytosolic components at low environmental pH seems to be clearly presented, but the role of the injectisome basal body in this process may need to be more critically evaluated. Below are some issues that the authors should address to make the work more complete.

We thank the reviewer for this constructive evaluation. We agree that the role of the basal body in the pH response is central, and have therefore performed additional experiments to address this point in the revised version. Perhaps most importantly, we now include new data that supports a key role of SctD in the reaction to low external pH (new Fig. 4 and Suppl. Fig. 13, see below).

MAJOR CONCERNS

1. In Suppl Fig 9, there is a clear co-localization of SctF with SctL under the initial pH 7 conditions, however, this colocalization at pH 4 and upon return to pH 7 appear to be very similar. In some other places the data seem to be open to multiple interpretations. For example, in Fig 2, the return to the original pH 7 state for SctL and SctN doesn't seem to be so clear cut though it does appear so for SctQ and SctK. These are challenging experiments to perform, so rigorous critical evaluation of the data is essential. The authors should discuss the potential difficulties in data analysis and interpretation.

This comment highlights several important aspects. (i) The co-localization of SctF and SctL indeed is difficult to quantify, as red fluorophores (which are needed in this experiment for the co-localization) bleach relatively quickly in time course series, especially when green fluorescence is measured in parallel. To improve this, we have repeated the experiment using EGFP-SctL and the red fluorescent dye CF 568 to label SctF_{55C}. The results are shown in what is now Suppl. Fig. 11. They more clearly document that both SctF and SctL localize at the same position before and after the pH 4 treatment, and that both proteins initially co-localize. Although this infers a colocalization after pH 4 treatment, we were not able to directly quantify this and therefore removed the respective direct statement (lines 154-157). (ii) Indeed, we also noted that the more proximal cytosolic components SctL and especially SctN appear to re-bind more slowly and/or less completely after a pH 4 treatment. This makes sense, given the chain of interactions SctD-SctK-SctQ-SctL-SctN. However, we are careful not to overinterpret this impression. The effect is difficult to quantify, because the cytosolic components have different stoichiometries. While the fluorescence quantification and the automated spot detection are very reliable, we would like to avoid applying them for quantitative comparisons across different components. In the revised manuscript, we now mention these points and clearly address these difficulties in data analysis and interpretation (lines 445-449).

2. In Suppl Fig 12, the data for *Shigella* are not quite as convincing as the authors state and the question that arises is whether or not there is a possibility that *Shigella* can actually invade the epithelium of the stomach. This and the fact that *Pseudomonas aeruginosa* actually appears to form more and better defined punctate foci warrants mentioning.

We have repeated the measurements for Fig. 4 and Suppl. Fig. 12 (now Fig. 6 and Suppl. Fig. 18), using the same medium for all three organisms (as suggested by reviewer 4, point 5). The results (and the bigger fields of view now displayed in Suppl. Fig. 18) confirm a significant decrease of the fraction of T3SS-positive *Shigella* at pH 4, as well as the lack of influence on *Pseudomonas*, although the fraction of T3SS-positive *Pseudomonas* is lower in BHI. Independently of these measurements, the number and contrast of fluorescent foci is indeed lower in *Shigella*, potentially due to the lower stoichiometry of SctN. We now explicitly mention these considerations in the results part (lines 315-317).

3. As the authors begin to propose how such events might be triggered, they home in on SctD and SctV. Several recent papers propose that SctV undergoes rearrangement in the absence of the injectisome cytosolic components (perhaps as a checkpoint to prevent premature effector secretion), however, the authors spend very little time discussing SctV and focus largely on SctD. In the absence of defining the residues of SctD that might be involved in disassembly of the injectisome cytosolic components, the authors should more critically consider the role of SctV in this process.

Indeed, SctV, with its central role in secretion and its regulation, would also be a candidate to contribute to the reaction to external pH. Given that the reaction is induced by external (extracellular or periplasmic) pH, we screened SctV for potential amino acids that could confer the observed phenotype of dissociation of the cytosolic components at low pH. We found few protonatable amino acids in the periplasmic SctV domains, none of which was a suitable candidate to confer the observed phenotype (amino acids that acquire a positive charge at low pH in *Yersinia* and that are neutral in *Pseudomonas*) (shown in the new Suppl. Table 1). In addition, the strong effect of SctD (but not SctV) overexpression on the dissociation at low external pH (new Fig. 4, see response to point 7) strongly suggests a central role of SctD, rather than SctV. This is further supported by the localization of SctV-EGFP at different pH (Fig. 5C and new Suppl. Fig. 17). However, the re-integration of SctV may play a role in the delayed re-activation of secretion, which we now discuss in lines 473-476.

4. For the SctD experiments, it is not clear that the EGFP-SctD fusion was tested to see if it retained its T3SS secretory activity or allowed assembly of the injectisome cytosolic components. This needs to be done or made more apparent that it was done. The EGFP probe appears to be at the SctD N terminus which would place it on the cytoplasmic domain of the protein which is the domain through which it interacts with SctK. This domain is small (~13kDa) and the mass of EGFP is on the order of 33kDa, which could sterically influence its interaction with SctK. Likewise, it's unclear how this added density would influence the packing of the inner membrane ring or even the interaction between SctD and SctJ or SctC. Any compromise of the IM ring stability should be discussed.

The EGFP-SctD fusion is stable and functional, albeit with a slightly reduced secretion efficiency. This was shown earlier (Diepold et al., PLOS Biol 2015, doi: 10.1371/journal.pbio.1002039); we also repeated the experiments with the settings used for this manuscript and present the results in the new Suppl. Fig. 15AB. As the presence of all cytosolic components is required for secretion, this result also suggests that the cytosolic components assemble normally. Nevertheless, we now mention a possible destabilization of the IM ring by the fusion and performed further experiments to test this possibility: we assayed the localization of EGFP-SctQ in a single-labeled strain and an EGFP-SctQ mCherry-SctD double labeled strain at different pH, based on the assumption that a destabilization of the IM ring by the fluorescent tag would also destabilize the cytosolic components. EGFP-SctQ localized in the same way in both strains at all tested pH (new Suppl. Fig. 15C), suggesting that the IM ring is not significantly destabilized by fluorescent labeling of SctD. We now describe these controls in the results section (lines 250-251, 266-270).

5. With respect to the model proposed for SctD dissociation from the injectisome, the authors should consider the following: SctD is a rather stable component of the injectisome that, unlike the

injectisome cytosolic components, remains associated with the complex even after somewhat harsh purification processes. Likewise, assembly of the IM ring has been proposed by multiple groups and in reviews as being an early step in overall injectisome assembly. For it to later disassociate from the injectisome seems somewhat counterintuitive. The authors should reconcile this assumption with their model.

As mentioned, we have added new data supporting the conclusion that indeed, SctD has a central role in the adaptation of the T3SS to low pH, and at least partially dissociates under these conditions. Stability in purified structures and an early role in assembly does not mean that a protein does not exchange with high dynamics, even in fully functioning machineries. This has been shown very prominently for the flagellum (nicely reviewed in Tusk et al, JMB 2018, doi: 10.1016/j.jmb.2018.06.039) and the DNA replisome (Mueller et al, Biophys Rev 2019, doi: 10.1007/s12551-019-00569-4), so uniform and stable structures and protein exchange are not mutually exclusive. Besides showing the new data (Fig. 4), we now discuss this topic in more detail (lines 455-461) in the hope to reconcile this apparent mismatch.

6. Related to the above comment, it appears in the proposed model that dispersal of SctD in the IM occurs via passive diffusion. Is this the case or is it a directed process as might be suggested by the ability to rapidly reassemble back into an active complex when the pH is neutralized? Furthermore, how might membrane fluidics play a role in dispersion and reassembly?

This is an interesting question. Indeed, we assume this process to occur via passive diffusion. While we cannot exclude an independent active mechanism, the values we measured for lateral diffusion (75-180 nm median jump distance in 200 ms intervals for the fast fraction in the membrane) correspond to diffusion coefficients of 0.006-0.048 $\mu\text{m}^2/\text{s}$, which is in the same range as measurements for freely diffusing membrane proteins with 4 TMHs in *E. coli* (Kumar et al, Biophys J, 2010; doi: 10.1016/j.bpj.2009.11.002), the membrane proteins LacS-GFP and BcaP-GFP in *Lactococcus lactis* (Mika et al, Mol Microbiol, 2014, doi: 10.1111/mmi.12800), cytochrome bd-I complexes in the *E. coli* plasma membrane (Lenn et al, Mol Microbiol, 2008; doi: 10.1111/j.1365-2958.2008.06486.x), and mobile GFP-MotB (Leake et al, Nature 2006, doi: 10.1038/nature05135). These values would allow a re-binding of SctD to the complex that is well within the measured recovery times of the foci, independently of the precise membrane fluidity. We thank the referee for raising this point, which we now also discuss in the revised manuscript (lines 450-454).

7. With respect to the mutational analysis of SctD, removal of key charged amino acids had no effect on SctQ dispersal, but the authors do comment that *Pseudomonas* and *Yersinia* SctD are relatively well conserved (Suppl Fig 12B). Thus, could the authors substitute the entire protein or the entire periplasmic domain to test their model? In doing so, the authors may find out that either the trigger is not SctD, but maybe SctJ (or that SctV may have a major role here). Likewise, this might help in determining whether the fused EGFP is partially destabilizing to the IM ring.

As suggested, we complemented a *Y. enterocolitica* ΔSctD EGFP-SctQ strain with the *P. aeruginosa* homolog *PaSctD* from plasmid. The complemented strain was functional for secretion (new Suppl. Fig. 13D). Strikingly, *in trans* overexpression of both proteins, *YeSctD* or *PaSctD*, strongly reduced the dissociation of EGFP-SctQ at low external pH (new Fig. 4 and Suppl. Fig. 13). These results strongly suggest that the increased levels of free SctD shift the dissociation equilibrium towards the bound form of SctD, which then allows the binding of the cytosolic components even at an external pH of 4. Overexpression of the other basal body components SctC, SctJ or SctV did not have any effect (new Fig. 4). These findings support a central role of SctD in the reaction of the T3SS to low external pH and indicate that differences in the local SctD concentration, rather than amino acid changes, are responsible for the differences in phenotype between species. We discuss these core additional findings in lines 404-424.

MINOR CONCERNS

1. The writing could use a little smoothing out (paragraph formats, etc.) and a bit more time spent discussing the results would be helpful.

As mentioned for the individual points, we have now significantly extended the discussion of our results, which in view also resulted in a clearer internal structure of the discussion.

2. Question: Is there evidence that YadA contributes to adherence to the mucosal surface in the stomach? In other words, is T3SS secretion really a concern physiologically in the stomach?

While our data shows that YadA adheres to abiotic surfaces and collagen at pH 4, there is no direct evidence for adherence to the mucosal surface in the stomach. We have therefore changed the discussion about the potential physiological role of the pH effect (lines 462-484) and now mention both the direct suppression of secretion at low external pH and a more efficient passage of the M cells due to the slightly delayed reactivation after pH 4 as two potential (but unproven) alternatives.

3. Related to the comment above, upon ingestion the temperature is elevated to 37°C at which time the T3SS begins to be expressed. Fig 6 suggests that full expression of the T3SS injectisome takes about 2 hours, but Fig 1B suggests that the bacteria pass through the gut in as little as 30 minutes and certainly by 4 hours. Based on this, do the authors consider accidental activation of the T3SS while in the stomach to be a problem the pathogen must overcome?

As shown in Fig. 6 and earlier (Diepold et al. 2010, doi: 10.1038/emboj.2010.84 and other groups), visible secretion of effectors into the supernatant starts about 60 minutes after the shift to 37°C. Host cell contact might activate secretion more quickly and even at temperatures below 37°C (Kusmieriek et al., PLOS Pathogens 2019, doi: 10.1371/journal.ppat.1007813). In both cases, activation in the stomach might be relevant. As mentioned above, while the effect is very clear, its relevance during infection is difficult to determine, and we have adapted the discussion accordingly.

Reviewer #2 (Remarks to the Author):

In this manuscript, the authors report on the finding that gastrointestinal pathogens with T3SS utilize pH change as a means of regulating the timing of effector protein secretion into a host cell. They demonstrate that *Yersinia enterocolitica* cultures remain viable after incubation at a pH equivalent to that of passage through the stomach and that exposure to this acidic environment inhibits the secretion of effector proteins without affecting the localization of T3SS needle components to specific foci at the bacterial membrane. The authors also showed that while secreted effector proteins do not colocalize with the needle complex at low pH, the secretion phenotype is restored upon reintroduction to physiological pH. They hypothesized that the periplasmic T3SS protein SctD was involved in external pH sensing and regulating effector secretion and found that whereas SctD did not colocalize with needle foci at low pH, changes in external pH did not alter the protonation state of SctD. Among the species they tested, this behavior was only observed in T3SS pathogens of the GI tract.

Overall, the data presented are interesting, the quality of the data is outstanding, and the data support the authors' conclusions. I have a few major concerns that should be addressed regarding controls that are needed to support their key findings.

We thank the reviewer for the detailed and positive evaluation and have addressed the individual points below.

Major concerns:

1. A critical missing control is whether the fluorescent signal per se is decreased by low pH. Indeed, in a paper from GH Patterson et al. (*Biophys J* 1997), the fluorescence of EGFP decreased more than 80-fold when the fluorophore was shifted from pH 7.0 to pH 4.0. The authors need to perform control experiments to test whether this is occurring here for each of the fluorophores they use (they could use a cytoplasmic EGFP and mCherry and quantify their signal intensities under conditions in which the cytoplasm is equilibrated to this range of pH's. Although the increase in cytosolic fluorescence at pH 4 reported by the authors in Fig. 2C would argue against pH-dependent decreases in fluorescence per se, the experiment shown in Fig. 2C is not sufficient as this control, as the two conditions being compared also differ in other important respects.

This is indeed an important aspect. Various studies, including the mentioned Patterson et al. paper and Doherty et al, *BMC Res Notes* 2010 (doi 10.1186/1756-0500-3-303) have shown that GFP is strongly pH-dependent, with a sharp decrease of fluorescence between pH 7 and pH 4, while mCherry fluorescence is less pH-dependent. Importantly, in our experiments, the fluorophores are in the cytosol, where the pH remains in a more neutral range at low external pH values (Fig. 3A). This is visible in our micrographs (which are all intensity-scaled identically within individual figures); however, to specifically measure the pH dependence of intracellular EGFP and mCherry at different external pH, we performed additional experiments. The results (shown in the new Suppl. Fig. 7) confirm that, while the extracellular pH has an influence on EGFP fluorescence, a drastic loss of fluorescence at low external pH is only observed in the presence of DNP (where intracellular = extracellular fluorescence). Also in accordance with published data, pH influence on mCherry is less pronounced.

2. Since *Yersinia pestis* bypasses the intestine, being instead introduced into the host by injection from fleas into the bloodstream, the author's model would predict that the *Y. pestis* SctN and SctQ homologs would differ from those of *Y. enterocolitica* and that localization in *Y. pestis* would be independent of pH. This seems likely to be a better comparator than *P. aeruginosa*. The authors should examine this and potentially residues in the SctD homologs that differ between the two

species and which might be protonated at pH 4 but not at pH 7. (For experiments, to avoid any biosafety issues, the vaccine strain of *Y. pestis* could be used.)

That's a good idea. However, as can be seen in a larger multiple sequence alignment that we now include for all core proteins of this study for the *Y. enterocolitica*, *Y. pestis*, *P. aeruginosa*, *S. flexneri* and SPI-2 homologs (new Suppl. Table 1), the components of the *Y. pestis* and *Y. enterocolitica* T3SS are very closely related (around 99% amino acid identity, with none of the differing amino acids being a candidate for changing their charge between pH 7 and 4). Given that a suppression of secretion at low pH would not be detrimental for *Y. pestis*, it is possible that the effect is conserved in *Y. pestis*, or suppressed by more subtle changes, which would be in line with our new findings suggesting that the local concentration of SctD is key for its function in pH sensing (see new Fig. 4 and Suppl. Fig. 13).

3. Statistics: for each set of experiments, the authors should indicate the number of independent experiments (e.g., Fig. 5B) and provide statistical significance appropriate based on independent replicates. In some cases, it is difficult to tell whether the data presented represent biological or technical replicates (e.g., Fig. 4).

We now provide these numbers and the statistics in the figure legends and/or the materials and methods part. Where necessary, we have performed additional experiments to include three or more independent biological replicates with all strains measured in the same experiment.

Minor comments:

1. Line 16: Please clarify "proliferate in contact to..."

To account for the inhibition of growth and division that accompanies T3SS activity in many bacteria, we have changed this sentence to "In order to increase survival chances in contact to eukaryotic host cells..."

2. Line 80: Figure 1B shows that *Yersinia* persists in the stomach for a minimum of 30 minutes. Explain why 15 minutes was used as the experimental condition.

Thank you for the comment. To be consistent, we have repeated the experiment with a 30 minute incubation at the different pH values, which yielded very similar results and is now shown in the new Fig. 1C.

3. Lines 82-83: Optical density does not equate to viability as cells could be dead but still intact and thereby contribute to the optical density of a culture. This suggests the possibility that by exposing the cultures to a low pH, the authors were selecting for a subpopulation of bacteria. Moreover, in Suppl. Fig. 1, there appears to be a fitness decrease, as half of the cells appear to die when exposed to a low pH, and these cells are slower to recover when returned to physiological pH. The authors should be less dogmatic about their conclusions that low pH had no fitness selection effect.

This is correct. We now describe the result more carefully in the figure legend and the results part (lines 84-88).

4. Please include statistical tests in Figure 1D

We have performed additional repetitions of the YadA binding assay, which showed a higher variation of both technical and experimental replicates. While in all cases, binding at all tested pH values was observed, the overall results (no statistically significant difference between binding at pH 4 and pH 7, but significantly higher binding at pH 5, 6, and 8) are difficult to interpret. We therefore

de-emphasized this point by moving the figure to the supplementary results (new Suppl. Fig. 2), and also interpreted the results more conservatively both in the results part (lines 90-92) and the discussion (lines 390-392).

5. Line 88: What general surfaces? Abiotic surfaces?

Yes, this includes abiotic surfaces, which we now specifically mention: “*Y. enterocolitica* adheres to host cells, other bacteria and abiotic surfaces by adhesins....”

6. Please add error bars on Suppl. Fig. 7.

We have included error bars in this figure (now Suppl. Fig. 9) and also present the distribution of spots per bacterium in the new Supplementary Table 2.

7. Figure 3A, right panel: error bar for 10 minutes at pH4 is not consistent with the rest.

Thanks for the notice, we have corrected this mistake.

8. Figure 4B, pH7: it is unclear whether the dynamics of SctD are T3SS specific or typical of all inner membrane proteins. A useful control might be a known static membrane protein.

The main peak for the measured mean jump distances (mjd) for SctD at pH 7 (mjd between 0 and 60 nm, with a peak between 15 and 45 μm) is indeed typical for a static membrane protein. This corresponds to the measurement uncertainty; as can be seen in the micrographs themselves, the corresponding tracks slightly “wobble” around a center, but do not display any directional movement. This is in line with earlier general considerations for localization precision of fixed molecules by Wieser and Schütz (Methods 2008, doi: 10.1016/j.ymeth.2008.06.010) and the corresponding calculated diffusion coefficient of less than 0.003 $\mu\text{m}^2/\text{s}$ is similar to measurements for proteins bound to the flagellum (Fukuoka et al, JMB 2007, doi: 10.1016/j.jmb.2007.01.015) – we now specifically mention this in lines 257-259.

Likewise, the measured dynamics for the mobile fraction of SctD at pH 4 (mjd of 75-180 nm, corresponding to a diffusion coefficient of 0.006 – 0.048 $\mu\text{m}^2/\text{s}$) is in line with measured diffusion coefficients for bacterial membrane proteins, such as the chemoreceptor Tar, LacY and MtlA (Kumar et al., Biophys J 2010, doi: 10.1016/j.bpj.2009.11.002) and mobile GFP-MotB, which is probably the most comparable example to mobile SctD (Leake et al, Nature 2006, doi: 10.1038/nature05135). This is discussed in the revised manuscript (lines 450-454).

9. Lines 276-277: Please clarify whether Shigella and Pseudomonas are expressing recombinant Yersinia SctQ or the Shigella and Pseudomonas homologs to SctQ.

In all cases, the respective native homologs were expressed. We now clarify this by adding the species acronym (e.g. “*S. flexneri* GFP-SfSctN” and “*P. aeruginosa* EGFP-PaSctQ” (lines 315-321).

10. Figure 5: Please include error bars and statistics.

We have now included error bars and statistics for the new version of the figure, for which we incubated all species in the same buffer, as suggested by reviewer 4 (point 5).

11. Line 295: “Showever”

We removed the word in the context of other changes in this paragraph.

12. Figure 6A: Clarify whether foci are present and effectors and needle components are being expressed at 28C.

In the figure legend (now Fig. 7A), we now specifically state that “...at 28°C, (...) expression of injectisome components and effectors is downregulated and no injectisomes are assembled.”

13. Line 351 and Figure 4: The authors show that SctD dissociates from foci at low external pH but that this dissociation was not due to differential protonation states of SctD. Given that there are other periplasmic components of the T3SS injectisome, it is also possible that SctD does not directly sense external pH and instead, another periplasmic protein undergoes conformational changes as a direct result of the drop in external pH resulting in dissociation of SctD from the complex. These possibilities (and others) should be discussed.

We have performed a series of additional experiments and found that higher expression levels of SctD (but not SctC, SctJ or SctV) strongly reduce the dissociation of the cytosolic T3SS components at low pH (new Fig. 4 and Suppl. Fig. 13). This finding strongly suggests that SctD is the key component in pH sensing. Whether the responsible interaction(s) occur at the SctD-SctD interface or between SctD and SctC or SctJ cannot be directly deduced from our data; however, direct sequence comparisons between *Yersinia*, *Shigella* and *Pseudomonas* did not provide any candidate amino acids in these proteins or SctV (new Suppl. Table 1), and the fact that overexpression of SctC or SctJ had no influence on the pH-dependent dissociation of the cytosolic components indicates that the SctD-SctD interface is responsible. We now discuss these possibilities and the new results in more detail (lines 405-427).

14. Figure 12B: The authors showed that *Pseudomonas* effector secretion via T3SS does not respond to low external pH as *Yersinia* does, but that *Yersinia* and *Shigella* both respond similarly to low external pH. It would be informative to compare the sequences of periplasmic T3SS proteins between *Yersinia* and *Shigella* and use the similarities as a cross reference against the differences found between *Yersinia* and *Pseudomonas*.

We have performed a multiple sequence alignment of several species, including *Y. enterocolitica*, *P. aeruginosa* and *S. enterica*, for the main proteins used in this study, including the periplasmic proteins (new Suppl. Table 1). We did not detect any candidate residue where *Yersinia* and *Shigella* have the same (protonatable) residue, whereas the *Pseudomonas* homolog differs. This may also be due to the fact that the sequence conservation between *Yersinia/Pseudomonas* and *Shigella* is rather low, except for the two most highly conserved proteins SctN and SctV. Notably, our finding that the local concentration of SctD determines its pH dependence (new Fig. 4) suggests that the different response of the various T3SS is not necessarily caused by direct protonation of amino acids.

15. Lines 391-392: Please clarify “dissociation of the cytosolic components occurs at a slightly lower pH than the loss of effector secretion”.

We now specify the pH ranges of dissociation (between 4 and 5) and loss of effector secretion (between 5 and 6).

16. Line 398: The authors carried out secretion experiments in BHI, which is dissimilar to the intracellular environment of an M cell, therefore cannot conclude that we will see the same secretion delay in an M cell as observed in BHI. Would be careful not to overstate the conclusions here.

This is absolutely correct, especially as studies have observed T3SS induction under conditions where low calcium levels are not sufficient (Kusmierik et al., PLOS Pathogens 2019, doi: 10.1371/journal.ppat.1007813). We now explicitly include this caveat in the discussion (lines 462-466).

17. Lines 406-407: Clarify whether the amino acid in SctV is susceptible to protonation changes.

We now specify the mutation (V632D). This amino acid is in a conserved region of SctV, which we highlight in the new Suppl. Table 1. Although protonation of the Asp residue in the mutated protein at low pH is conceivable, it is unclear how this explains the observed phenotype in the reference (loss

of interaction between SctV and SctW). Non-protonatable residues are conserved at this position in all tested species (Suppl. Table 1).

18. Line 497: “we fluorescent foci”

Corrected, thank you.

Reviewer #3 (Remarks to the Author):

Wimmi et al. present a thorough study on the pH-dependent machinery for bacterial injection of molecular toxins into host cells. The study reveals a molecular mechanism for the dissociation of the cytosolic T3SS components in response to extracellular pH variations and thus to environmental control of the bacterial injection of molecular toxins. The study resolves the contribution of various T3SS protein components to the pH-dependent disassembly of the T3SS complex. Fluorescence microscopy was used to monitor the dissociation of the T3SS complex. Single-molecule tracking was employed to observe changes in the molecular mobility of SctD and reveal its role as modulator for pH sensing. Observations for various environmental conditions and bacterial pathogens confirmed that the observed mechanism is of general relevance.

In my opinion, this is an excellent manuscript. The study is well designed and of superb quality with regard to experimental work, data analysis and data interpretation. The fluorescence microscopy data is of high quality. Quantitative analysis is carried out in great detail and all statistical information is provided. The analysis allows for great confidence on the correct interpretation of the presented data. Without any hesitation I like to recommend publication of the manuscript in Nature Communications.

We thank the reviewer for this positive evaluation!

Reviewer #4 (Remarks to the Author):

The manuscript by Wimmi et al describes a phenomenon whereby several cytosolic injectisome components appear to dissociate from the inner membrane component SctD/SctD upon shift to low pH. This dissociation leads to a dramatic reduction in secretion of T3SS components and is reversible which allows the bacteria to rapidly commence secretion at high pH. The authors propose that this observation provides a means of preventing secretion of Yops in the stomach after oral ingestion of *Yersinia* (and *Shigella*), but rapidly ramping up secretion as the bacteria experience a rise in pH towards the distal end of the small bowel. The idea is intriguing and the authors data with various forms of microscopy using fluorescent probes support their model. Much of the data is rigorously controlled with ample data sets taken from 2 or more experiments, although in some panels it is unclear how many experiments were performed and how many cells evaluated for the conclusions drawn (for instance, 2D, 2E, 3B, 4C). The paper is generally well written for the broad readership of Nature Communication and the findings are of interest to the audience. Several weaknesses are noted in points below:

We thank the referee for the positive and constructive evaluation and have addressed the individual points as stated below. We now also include the number of experiments and evaluated cells for all figures.

1. The mechanism that controls the low pH response is not identified in the manuscript. The authors propose it is SctD since SctD also disperses upon low pH, but the evidence provided that shows SctD senses pH is missing. The authors show that in *Yersinia*, SctD become diffuse in membranes in low pH, while in *Pseudomonas*, SctD's 'diffusion' in membranes is the same regardless of pH (and intermediate between that found in *Yersinia* at pH7 versus pH 4), but this does not pin down *Yersinia* SctD as the molecule that senses low pH.

This is correct, and we performed a series of additional experiments to narrow down the mechanism. We reasoned that if pH-induced partial dissociation of the "sensor" protein induces the dissociation of the cytosolic complex, overexpression of this protein would shift the equilibrium towards the associated form of the protein itself and, in consequence, decrease dissociation of the cytosolic components at low pH. We found that overexpression of SctD strongly reduces the dissolution of foci of the cytosolic components at pH 4 (new Fig. 4 and Suppl. Fig. 13). In contrast, overexpression of any other candidate protein (SctC, SctJ, and SctV) did not reduce the dissociation of the cytosolic components (Fig. 4). This strongly supports the hypothesis that SctD plays a central role in pH sensing.

2. The authors make 4 amino acid substitutions of SctD in histidine, but these do not have a phenotype, so this also does not pin down the component sensing low pH. However, there are about 18 His, Glu and Asp in the periplasmic portion of SctD that are not found in *Pseudomonas*. Swapping out groups of these to support or eliminate their hypothesis is possible (or substituting the periplasmic portion of the *Pseudomonas* SctD for the *Yersinia* and determining if secretion is occurring, if these components are interchangeable as some components are). While the homology is very weak, can the authors try to suss out whether a subset of the potential H, D and Es are structurally conserved between *Shigella* and *Yersinia* but not *Pseudomonas* and substitute these?

As mentioned by the reviewer, the simplest model for a pH-dependent interaction switch would involve pH-dependent protonation of Asp, Glu or His. The phenotype observed in this study (dissociation at low pH in *Y. enterocolitica*, constant association in *P. aeruginosa*) can occur for an amino acid that acquires a positive charge at low pH in *Yersinia* and that is neutral in *Pseudomonas*, which is only the case for His residues in *Yersinia*, which are replaced by neutral amino acids in

Pseudomonas. Replacement of Asp or Glu by neutral amino acids in *Pseudomonas* would not lead to this phenotype (unless more complicated charge transfer arrangements are considered). Nevertheless, we performed additional experiments, including the suggested complementation of *Yersinia* Δ SctD with *Pa*SctD. Interestingly, higher expression levels of both *Ye*SctD and *Pa*SctD strongly reduced the pH-dependent dissociation of cytosolic T3SS components (Fig. 4, see reply to point 1), while at intermediate *Ye*SctD and *Pa*SctD levels, we still observe the pH effect (new Suppl. Fig. 13). Taken together, these data suggest that relative expression levels of SctD, rather than amino acid changes, confer the observed difference in phenotype.

3. Some of their reassembly data buried in the supplemental materials (figures 8-9, especially Suppl Fig 9) were potentially quite intriguing and could have been highlighted more if shown in a rigorous manner. Information about the number of times these images were evaluated is missing, so it is unclear how strong these data and the points the authors are trying to make in the absence of rigorous evaluation. For instance, in the image in Suppl Fig 9 at pH4, mCherry-SctL does seem to colocalize even at pH4 and not much more is associated after re-shift to pH 7.

Thank you. We have extended our analysis of the dissociation and re-association kinetics, which we now present in a comparable and quantitative manner (previous Suppl. Fig. 7 and 8, now Suppl. Fig. 9 and 10). The experiment for the previous Suppl. Fig. 9 (now Suppl. Fig. 11) indeed is somewhat difficult to perform and analyze, as red fluorophores (needed in this experiment for the co-localization) bleach relatively quickly in time course series, especially when green fluorescence is measured in parallel. To improve this result, we have repeated the experiment using EGFP-SctL and the red fluorescent dye CF 568 to stain SctF_{55C}. The results in what is now Suppl. Fig. 11 more clearly show that both SctF and SctL localize at the same position before and after the pH 4 treatment, and that both proteins initially co-localize. Although this infers a colocalization after pH 4 treatment, we were not able to directly quantify this and therefore removed the respective direct statement. As requested, we now include the number of independent experiments for all figures and supplementary figures in the respective legends and/or the Material and Methods section.

4. All of the dissociation/diffusion is measured by various fluorescence techniques. An independent method for instance, crosslinking under both conditions or IPs, would strengthen these findings.

This is absolutely correct, and we have confirmed new fluorescence microscopy-based findings by co-IP for many strains, including the EGFP-SctQ, EGFP-SctD, EGFP-SctK and EGFP-SctL strains used in this manuscript, before, where possible (e.g. Diepold et al., EMBO J 2010, doi: 10.1038/emboj.2010.84 and Diepold et al., Nature Commun 2017, doi: 10.1038/ncomms15940). The very situation investigated in this study (low external pH and near-neutral cytosolic pH) makes it very hard to perform equivalent co-IP tests, as this pH gradient breaks down upon cell lysis, influencing any interactions. Likewise, we did not succeed to obtain reliable results by external crosslinking (either at pH 7 or pH 4), notwithstanding potential differences in crosslinking activity at the different pH. We think that the used fluorescence microscopy-based approach is the method of choice for this research question and have performed a high number of internal and dedicated control experiments (e.g. Suppl. Fig. 3, 4, 6, 7, 12, 14, 17) to support our findings.

5. Virtually all of their findings show that at low pH, T3SS components disaggregate with the exception of the SctF, but that molecule is external and we do not know whether the needles are still largely intact or if in fact they have mostly fallen apart and the probe is detecting SctF associated with SctC. A control is needed showing that low pH is not causing significant disruption of most membrane and membrane associated proteins in *Yersinia*. The best control is the *Pseudomonas* Suppl Fig 12, but these bacteria are grown in different media.

As suggested, we have performed the experiments for Suppl. Fig. 12 (comparison of T3SS in *Yersinia*, *Pseudomonas* and *Shigella*, now Suppl. Fig. 18) in the same medium (BHI) for all organisms. Although in this medium, the fraction of T3SS-positive *Pseudomonas* is slightly reduced under both conditions (pH 7 and pH 4), the results are comparable. To more directly show that an external pH of 4 does not generally disrupt membrane complexes, also we tested the assembly and localization of SctV-EGFP at the different external pH values. SctV-EGFP still forms fluorescent foci, most likely corresponding to nonamers (Abrusci et al., NSMB 2013, doi: 10.1038/nsmb.2452), at an external pH of 4 (overlay in Fig. 5C, single images in new Suppl. Fig. 17). Due to the opening of the SctD ring, these foci are mobile within the IM at low pH (similar to a delta-SctD strain, see Diepold et al., Mol Microbiol 2011, doi: 10.1111/j.1365-2958.2011.07830.x), an additional confirmation for the central role of SctD in the pH response.

6. The authors spend some time justifying their experiments by proposing that this mechanism is needed to prevent *Yersinia* from injecting Yops into cells in the stomach. But do cells lining the stomach express beta-1 integrins - the integrins that *Yersinia* primarily uses to attach to cells? Most cells in the intestine do not express beta-1 integrins – and it has been shown in mice at least that *Yersinia* bind to a very small population of cells in the intestine. Is there any evidence that *Yersinia* is more likely to invade stomach lining in mice treated with a proton pump inhibitors? (The data showing binding is in Fig 1D is not directly showing that attachment to cells occurs at pH 4.) Certainly, a more nuanced discussion of this rationale is required.

We agree that while our data shows that YadA adheres to abiotic surfaces and collagen at pH 4, there is no direct evidence for adherence to the mucosal surface in the stomach. We have therefore adapted the discussion about the potential physiological role of the pH effect and now discuss both the direct suppression of secretion at low external pH and a more efficient passage of the M cells due to the slightly delayed reactivation after pH 4 as two potential (but unproven) alternatives (lines 450-472). While our new finding that overexpression of SctD strongly reduces dissociation of the cytosolic T3SS components at low pH might open a path for mouse experiments to further investigate this point, these experiments are beyond the scope of this study.

Minor points:

1. Line 22 – more up to date references/reviews on *Yersinia* effector functions would be good – one of these, although wonderful is 22 years old and the other 13 years old. The field has made some advances since then.

Touché – in the introduction and discussion, we now include more up-to-date reviews for *Yersinia* and other animal pathogenic bacteria (Pha, 2016; Johnson *et al.*, 2018; de Jong and Alto, 2018; Shenoy *et al.*, 2018; De Souza Santos and Orth, 2019), as well as T3SS-plant interactions (Tampakaki *et al.*, 2014) and a discussion of diversity and evolution of T3SS effectors (Bastedo *et al.*, 2019).

2. Line 62 – more up to date references on translocation.

We added two more recent articles and one review that more specifically investigate translocation and host cell specificity (Armentrout and Rietsch, 2016; Nauth *et al.*, 2018; Bohn *et al.*, 2019).

3. Lines 137-139: Suppl Fig 6 seems to show that the dissociation kinetics are similar between secreting and non-secreting conditions for the proteins shown, not that it is reversible as described in lines 133-135 and shown in Supple Fig 5

Thanks for noticing, we corrected the statement.

4. Line 295 – change Showever to However

We removed “(S)however” in the context of other changes in the paragraph.

5. Suppl Fig 6. The legend states that rows 1 and 3 are secreting and rows 2 and 4 are non-secreting, but the figure shows otherwise.

Thanks for the thorough examination – the legend is right, and we have corrected the labels in the figure (now Suppl. Fig. 8).

6. I find giving all these components from different bacteria, the exact same name from when they function differently and often have poor sequence homology to each other, as the authors describe here functionally with the *Pseudomonas* system versus the *Yersinia/Shigella* system (or the *Shigella* versus *Yersinia/Pseudomonas* genes), problematic. While it does simplify remembering them all, it adds confusion when writing. Perhaps calling them SctDy versus SctDp for *Yersinia* version versus *Pseudomonas* would help resolve the conundrum.

This is a good point. When specifically referring to proteins from a certain species, we now add the initials in italics as a prefix (e.g. *YeSctD* or *PaSctN*), as suggested in Wagner and Diepold, 2020.

REVIEWERS' COMMENTS

Reviewer #1 (Remarks to the Author):

The authors have adequately addressed all of my concerns.

Reviewer #2 (Remarks to the Author):

The authors have done a nice job of responding to the reviewers' concerns. I have no additional concerns.

Reviewer #4 (Remarks to the Author):

The revisions by Wimmi et al strengthen the premise and findings of the original manuscript. The data are more robust, the presentation is clearer, and conclusions are more nuanced. The number of repeats and statistical analyses are stated. The authors have addressed all my major concerns.

Minor point:

The findings with YadA are interesting, however, Invasin has also been shown to play a major role in *Yersinia enterocolitica* particularly during penetration of epithelial cells in animal infections (see Han and Miller, 1997 and Pepe and Miller, 1993, and 1995). They may want to include invasin as a player in epithelial cell invasion.

Why does the binding dip at pH7 in Suppl. Fig 2?

Reviewer #1 (Remarks to the Author):

The authors have adequately addressed all of my concerns.

Reviewer #2 (Remarks to the Author):

The authors have done a nice job of responding to the reviewers' concerns. I have no additional concerns.

Reviewer #4 (Remarks to the Author):

The revisions by Wimmi et al strengthen the premise and findings of the original manuscript. The data are more robust, the presentation is clearer, and conclusions are more nuanced. The number of repeats and statistical analyses are stated. The authors have addressed all my major concerns.

Minor point:

The findings with YadA are interesting, however, Invasin has also been shown to play a major role in *Yersinia enterocolitica* particularly during penetration of epithelial cells in animal infections (see Han and Miller, 1997 and Pepe and Miller, 1993, and 1995). They may want to include invasin as a player in epithelial cell invasion.

Why does the binding dip at pH7 in Suppl. Fig 2?

Thanks for the comment. The role of invasin (whose expression is downregulated at 37°, but less so at low pH) indeed warrants its inclusion. We now mention the possible role of invasin in host cell contact during infection, both in the introduction (lines 53-55 in the manuscript version with tracked changes) and the discussion (lines 470-471) and refer to the studies mentioned by the reviewer, as well as additional papers.

We think that lower and less coherent adhesion at pH 7 is most likely caused by precipitation of YadA close to its isoelectric point ($pI_{YadA} = 7.2$), which we now also mention in the figure legend of Suppl. Fig. 2.